# Differentially Private Optimization with Sparse Gradients

**Badih Ghazi**
Google Research
badihghazi@google.com

**Cristóbal Guzmán**
Google Research and Pontificia Universidad Católica de Chile
crguzman@google.com

**Pritish Kamath**
Google Research
pritishk@google.com

**Ravi Kumar**
Google Research
ravi.k53@gmail.com

**Pasin Manurangsi**
Google Research
pasin@google.com

## Abstract

Motivated by applications of large embedding models, we study differentially private (DP) optimization problems under sparsity of *individual* gradients. We start with new near-optimal bounds for the classic mean estimation problem but with sparse data, improving upon existing algorithms particularly for the high-dimensional regime. The corresponding lower bounds are based on a novel block-diagonal construction that is combined with existing DP mean estimation lower bounds. Next, we obtain pure- and approximate-DP algorithms with almost optimal rates for stochastic convex optimization with sparse gradients; the former represents the first nearly dimension-independent rates for this problem. Furthermore, by introducing novel analyses of bias reduction in mean estimation and randomly-stopped biased SGD we obtain nearly dimension-independent rates for near-stationary points for the empirical risk in nonconvex settings under approximate-DP.

## 1 Introduction

The pervasiveness of personally sensitive data in machine learning applications (e.g., advertising, public policy, and healthcare) has led to the major concern of protecting users' data from their exposure. When releasing or deploying these trained models, differential privacy (DP) offers a rigorous and quantifiable guarantee on the privacy exposure risk [1].

Consider neural networks whose inputs have categorical features with large vocabularies. These features can be modeled using embedding tables; namely, for a feature that takes $K$ distinct values, we create trainable parameters $w_1, \ldots, w_K \in \mathbb{R}^k$, and use $w_a$ as input to the neural network when the corresponding input feature is $a$. A natural outcome of such models is that the per-example gradients are guaranteed to be sparse; when the input feature is $a$, then only the gradient with respect to $w_a$ is non-zero. Given the prevalence of sparse gradients in practical deep learning applications, GPUs/TPUs that are optimized to leverage gradient sparsity are commercially offered and widely used in industry [2, 3, 4, 5]. To leverage gradient sparsity, recent practical work has considered DP stochastic optimization with *sparse gradients* for large embedding models for different applications including recommendation systems, natural language processing, and ads modeling [6, 7].

Despite its relevance and promising empirical results, there is limited understanding of the theoretical limits of DP learning under gradient sparsity. This gap motivates our work.

38th Conference on Neural Information Processing Systems (NeurIPS 2024).

| Setting | Upper bound | | Lower bound | |
|---|---|---|---|---|
| $\varepsilon$-DP | $1 \wedge \sqrt{\frac{s \ln d}{\varepsilon n}} \wedge \frac{\sqrt{sd}}{\varepsilon n}$ | (Thm. 3.2) | $1 \wedge \sqrt{\frac{s \ln(d/(\varepsilon n))}{\varepsilon n}} \wedge \frac{\sqrt{sd}}{\varepsilon n}$ | (Thm. 4.1) |
| $(\varepsilon, \delta)$-DP | $1 \wedge \frac{(s \ln(d/s) \ln(1/\delta))^{1/4}}{\sqrt{\varepsilon n}} \wedge \frac{\sqrt{d \ln(1/\delta)}}{\varepsilon n}$ | (Thm. B.1) | $1 \wedge \frac{(s \ln(1/\delta))^{1/4}}{\sqrt{\varepsilon n}} \wedge \frac{\sqrt{d \ln(1/\delta)}}{\varepsilon n}$ | (Thm. 4.5) |

Table 1: Rates for DP mean estimation with sparse data of unit $\ell_2$-norm. Bounds stated for constant success/failure probability, resp. We use $a \wedge b$ to denote $\min(a, b)$. New results highlighted .

| Setting | Guarantee | New Upper bound (sparse) | | Upper bound (non-sparse) |
|---|---|---|---|---|
| $(\varepsilon, \delta)$-DP | Convex ERM | $\frac{(s \ln(d) \ln(1/\delta))^{1/4}}{\sqrt{\varepsilon n}} \wedge \mathcal{R}_{\varepsilon,\delta}$ | (Thm. 5.4, 6.1) | $\mathcal{R}_{\varepsilon,\delta}$ |
| | SCO | $\frac{(s \ln(d) \ln(1/\delta))^{1/4}}{\sqrt{\varepsilon n}} \wedge \mathcal{R}_{\varepsilon,\delta} + \frac{1}{\sqrt{n}}$ | (Thm. 6.3) | $\mathcal{R}_{\varepsilon,\delta} + \frac{1}{\sqrt{n}}$ |
| $\varepsilon$-DP | Convex ERM | $\left(\frac{s \ln(d)}{\varepsilon n}\right)^{1/3} \wedge \mathcal{R}_{\varepsilon}$ | (Thm. 6.1, G.4) | $\mathcal{R}_{\varepsilon}$ |
| | SCO | $\left(\frac{s \ln(d)}{\varepsilon n}\right)^{1/3} \wedge \mathcal{R}_{\varepsilon} + \frac{1}{\sqrt{n}}$ | (Thm. 6.3) | $\mathcal{R}_{\varepsilon} + \frac{1}{\sqrt{n}}$ |
| $(\varepsilon, \delta)$-DP | Emp. Grad. Norm | $\frac{(s \ln(d/s) \ln^3(1/\delta))^{1/8}}{(\varepsilon n)^{1/4}} \wedge \left(\mathcal{R}_{\varepsilon,\delta}\right)^{2/3}$ | (Thm. 5.4) | $\left(\mathcal{R}_{\varepsilon,\delta}\right)^{2/3}$ |

Table 2: Rates for DP optimization with sparse gradients, compared to best-existing upper bounds in the non-sparse case. In the above, the bounds are stated for constant success probability, the function parameters and polylog($n$) factors are omitted, $\mathcal{R}_{\varepsilon,\delta} = \sqrt{d \ln(1/\delta)}/(\varepsilon n)$, $\mathcal{R}_{\varepsilon} = d/(\varepsilon n)$, and our improvements are highlighted .

## 1.1 Our Results

We initiate the study of DP optimization under gradient sparsity. More precisely, we consider a stochastic optimization (SO) problem, $\min\{F_{\mathcal{D}}(x) : x \in \mathcal{X}\}$, where $\mathcal{X} \subseteq \mathbb{R}^d$ is a convex set, and $F_{\mathcal{D}}(x) = \mathbb{E}_{z \sim \mathcal{D}}[f(x, z)]$, with $f(\cdot, z)$ enjoying some regularity properties, and $\mathcal{D}$ is a probability measure supported on a set $\mathcal{Z}$. Our main assumption is gradient sparsity: for an integer $0 \leq s \leq d$,

$$\forall x \in \mathcal{X}, z \in \mathcal{Z} : \qquad \|\nabla f(x, z)\|_0 \leq s,$$

where $\|y\|_0$ denotes the number of nonzero entries of $y$. We also study empirical risk minimization (ERM), where given a dataset $S = (z_1, \ldots, z_n)$ we aim to minimize $F_S(x) := \frac{1}{n} \sum_{i \in [n]} f(x, z_i)$.

Our results unearth three regimes of accuracy rates for the above setting: (i) the small dataset size regime where the optimal rate is constant, (ii) the large dataset size where the optimal rates are polynomial in the dimension, and (iii) an intermediate dataset size regime characterized by a new high-dimensional rate[1] (see Table 1 and Table 2, for precise rates). These results imply in particular that even for high-dimensional models, this problem is tractable under gradient sparsity. Without sparsity, these polylogarithmic rates is impossible due to known lower bounds [8].

In Section 3, we start with the fundamental task of $\ell_2$-mean estimation with sparse data (which reduces to ERM with sparse linear losses [8]). Here, we obtain new upper bounds (see Table 1). These rates are obtained by adapting the projection mechanism [9], with a convex relaxation that makes our algorithms efficient. Note that for pure-DP, even our large dataset rate of $\sqrt{sd}/(\varepsilon n)$ can be substantially smaller than the dense pure-DP rate of $d/(\varepsilon n)$ [8], whenever $s \ll d$. For approximate-DP we also obtain a sharper upper bound by solving an $\ell_1$-regression problem of a noisy projection of the empirical mean over a random subspace. Its analysis combines ideas from compressed sensing [10] with sparse approximation via the Approximate Carathéodory Theorem [11].

In Section 4, we prove lower bounds that show the near-optimality of our algorithms. For pure-DP, we obtain a new lower bound of $\Omega(s \log(d/s)/(n\varepsilon))$, which is based on a packing of sparse vectors.

---

[1]We will generally refer to high-dimensional or nearly dimension-independent rates indistinguishably, meaning more precisely that the rates scale polylogarithmically with the dimension.

While this lower bound looks weaker than the standard $\Omega(d/(n\varepsilon))$ lower bound based on dense packings [12, 8], we design a novel bootstrapping via a block diagonal construction where each block contains a sparse lower bound as above. This, together with a padding argument [8], yields lower bounds for the three regimes of interest. For approximate-DP, we also use the block diagonal bootstrapping, where this time the blocks use classical fingerprinting codes in dimension $s$ [8, 13]. Our approximate-DP lower bounds, however, have a gap of $\ln(d/s)^{1/4}$ in the high-dimensional regime; we conjecture that the aforementioned compressed sensing-based upper bound is tight.

In Section 5, we study DP-ERM with sparse gradients, under approximate-DP. We propose the use of stochastic gradient (SGD) with a mean estimation gradient oracle based on the results in Section 3. This technique yields nearly-tight bounds in the convex case (similar to first row of Table 2), and for the nonconvex case the stationarity rates are nearly dimension independent (last row of Table 2). The main challenge here is the *bias in mean estimation*, which dramatically deteriorates the rates of SGD. Hence we propose a bias reduction method inspired by the simulation literature [14]. This technique uses a random batch size in an exponentially increasing schedule and a telescopic estimator of the gradient which—used in conjunction with our DP mean estimation methods—provides a stochastic first-order oracle that attains bias similar to the one of a full-batch algorithm, with moderately bounded variance. Note that using the full-batch in this case would lead to polynomially weaker rates; in turn, our method leverages the batch randomization to conduct a more careful privacy accounting based on subsampling and the fully-adaptive properties of DP [15]. The introduction of random batch sizes and the random evolution of the privacy budget leads to various challenges in analyzing the performance of SGD. First, we analyze a *randomly stopped method*, where the stopping time dictated by the privacy budget. Noting that the standard SGD analysis bounds the cumulative regret, which is a submartingale, we carry out this analysis by integrating ideas from submartingales and stopping times [16]. Second, this analysis only yields the desired rates *with constant probability*. Towards high probability results, we leverage a private model selection [17] based on multiple runs of randomly-stopped SGD that exponentially boosts the success probability (details in Appendix F).

In Section 6, we study further DP-SO and DP-ERM algorithms for the convex case. Our algorithms are based on regularized output perturbation with an $\ell_\infty$ projection post-processing step. While this projection step is rather unusual, its role is clear from the analysis: it leverages the $\ell_\infty$ bounds of noise addition, which in conjunction with convexity provides an error guarantee that also leverages the gradient sparsity. This algorithm is nearly-optimal for approximate-DP. For pure-DP, the previous algorithm requires an additional smoothness assumption, hence we propose a second algorithm based on the exponential mechanism [18] run over a net of suitably sparse vectors. Neither of the pure-DP algorithms matches the lower bound for mean estimation (the gap in the exponent of the rate is of $1/6$), but they attain the first nearly dimension-independent rates for this problem.

## 1.2 Related Work

DP optimization is an extensively studied topic for over a decade (see [8, 19, 20], and the references therein). In this field, some works have highlighted the role of *model sparsity* (e.g., using sparsity-promoting $\ell_1$-ball constraints) in near-dimension independent excess-risk rates for DP optimization, both for ERM and SCO [21, 22, 23, 24, 25, 26, 27]. These settings are unrelated to ours, as sparse predictors are typically related to dense gradients.

Another proposed assumption to mitigate the impact of dimension in DP learning is that gradients lie (approximately) in a low dimensional subspace [28, 29, 30, 31] or where dimension is substituted by a bound on the trace of the Hessian of the loss [32]. These useful results are unfortunately not applicable to our setting of interest, as we are interested in arbitrary gradient sparsity patterns for different datapoints.

Substantially less studied is the role of gradient sparsity. Closely related to our work, [6] studied approximate DP-ERM under gradient sparsity, with some stronger assumptions. Aside from an additional $\ell_\infty$ bound on individual gradients, the following *partitioning sparsity assumption* is imposed. The dataset $S$ can be uniformly partitioned into subsets $S_1, \ldots, S_m$ with a uniform gradient sparsity bound: for all $k \in [m]$ and $x \in \mathcal{X}$, $\left\| \sum_{z \in S_k} \nabla f(x, z) \right\|_0 \leq c_1$. The work shows polylogarithmic in the dimension rates, for both convex and nonconvex settings. Our results only assume individual gradient sparsity, so on top of being more general, they are also faster and provably nearly optimal in the convex case. Another relevant work is [7], which studies the computational and utility benefits for DP with sparse gradients in neural networks with embedding tables. With the

caveat that variable selection on stochastic gradients is performed at the level of *contributing buckets* (i.e., rows of the embedding table), rather than on gradient coordinates, this work shows substantial improvements on computational efficiency and also on the resulting utility.

In [33], bias reduction is used to mitigate the regularization bias in SCO. While they also borrow inspiration from [14], both their techniques and scope are unrelated to ours.

## 1.3 Future Directions

We present some of the main open questions and future directions of this work. First, we conjecture that for approximate-DP mean estimation—similarly to the pure-DP case—a lower bound $\Omega\big(\sqrt{s\log(d/s)\ln(1/\delta)}/[n\varepsilon]\big)$ should exist; such construction could be bootstrapped with a block-diagonal dataset for a tight lower bound (Lemma 4.3). Second, for pure DP-SCO, we believe an algorithm should exist that achieves rates analogous to those for mean estimation. Unfortunately, most of variants of output perturbation (including phasing [20, 24, 34]) cannot attain such rates. From a practical perspective, the main open question is whether our rates are attainable without prior knowledge of $s$; note that all our mean estimation algorithms (which carries over to our optimization results) depend crucially on knowledge of this parameter. While we can treat $s$ as a hyperparameter, it would be highly beneficial to design algorithms that automatically adapt to it.

We believe our bias reduction is of broader interest. For example, [35, 36] have shown strong negative results about bias in DP mean estimation. While similar lower bounds may hold for sparse estimation, bias reduction allows us to amortize this error within an iterative method, preventing error accumulation.

Finally, there is no evidence of our nonconvex rate being optimal. In this vein, we should remark that even in the dense case the optimal stationarity rates are still open [37].

## 2 Notation and Preliminaries

In this work, $\|\cdot\| = \|\cdot\|_2$ is the standard Euclidean norm on $\mathbb{R}^d$. We will also make use of $\ell_p$-norms, where $\|x\|_p := \big(\sum_{j\in[d]}|x_j|^p\big)^{1/p}$ for $1 \le p \le \infty$. For $p = 0$, we use the notation $\|x\|_0 = |\{j \in [d] : x_j \ne 0\}|$, i.e., the size of the support of $x$. We denote the $r$-radius ball centered at $x$ of the $p$-norm in $\mathbb{R}^d$ by $\mathcal{B}_p^d(x, r) := \{y \in \mathbb{R}^d : \|y - x\|_p \le r\}$. Given $s \in [d]$ and $L > 0$, the set of *s-sparse vectors* is (the scaling factor $L$ is omitted in the notation for brevity)

$$\mathcal{S}_s^d := \{x \in \mathbb{R}^d : \|x\|_0 \le s, \|x\|_2 \le L\}. \tag{1}$$

Note that Jensen's inequality implies: if $\|x\|_0 \le s$ and $1 \le p < q \le \infty$, then $\|x\|_p \le s^{1/p-1/q}\|x\|_q$.

**Remark 2.1.** *The upper bound results in this paper hold even if we replace the set $\mathcal{S}_s^d$ of sparse vectors by the strictly larger $\ell_1$-ball $\mathcal{B}_1^d(0, L\sqrt{s})$. Note that while our upper bounds extend to the $\ell_1$ assumption above, our lower bounds work under the original sparsity assumption.*

Let $f : \mathcal{X} \times \mathcal{Z} \mapsto \mathbb{R}$ be a loss function. The function evaluation $f(x, z)$ represents the loss incurred by hypothesis $x \in \mathcal{X}$ on datapoint $z \in \mathcal{Z}$. In *stochastic optimization* (SO), we consider a data distribution $\mathcal{D}$, and our goal is to minimize the expected loss under this distribution

$$\min_{x\in\mathcal{X}} \Big\{ F_{\mathcal{D}}(x) := \mathbb{E}_{z\sim\mathcal{D}}[f(x, z)] \Big\}. \tag{SO}$$

Throughout, we use $x^*(\mathcal{D})$ to denote an optimal solution to (SO), which we assume exists. In the *empirical risk minimization* (ERM) problem, we consider sample datapoints $S = (z_1, \ldots, z_n)$ and our goal is to minimize the empirical error with respect to the sample

$$\min_{x\in\mathcal{X}} \Big\{ F_S(x) := \tfrac{1}{n}\sum_{i\in[n]} f(x, z_i) \Big\}. \tag{ERM}$$

We denote by $x^*(S)$ an arbitrary optimal solution to (ERM), which we assume exists. Even when $S$ is drawn i.i.d. from $\mathcal{D}$, solutions (or optimal values) of (SO) and (ERM) do not necessarily coincide.

We present the definition of differential privacy (DP), deferring useful properties and examples to Appendix A. Let $\mathcal{Z}$ be a sample space, and $\mathcal{X}$ an output space. A dataset is a tuple $S \in \mathcal{Z}^n$, and datasets $S, S' \in \mathcal{Z}^n$ are *neighbors* (denoted as $S \simeq S'$) if they differ in only one of their entries.

**Definition 2.2** (Differential Privacy)**.** Let $\mathcal{A} : \mathcal{Z}^n \mapsto \mathcal{X}$. We say that $\mathcal{A}$ is $(\varepsilon, \delta)$-*(approximately) differentially private (DP)* if for every pair $S \simeq S'$, we have for all $\mathcal{E} \subseteq \mathcal{X}$ that $\Pr[\mathcal{A}(S) \in \mathcal{E}] \le e^\varepsilon \cdot \Pr[\mathcal{A}(S') \in \mathcal{E}] + \delta$. When $\delta = 0$, we say that $\mathcal{A}$ is $\varepsilon$-DP or pure-DP.

## 3 Upper Bounds for DP Mean Estimation with Sparse Data

We first study DP mean estimation with sparse data. Our first result is that the projection mechanism [9] is nearly optimal, both for pure- and approximate-DP. In our case, we interpret the marginals on each of the $d$ dimensions as the queries of interest: this way, the $\ell_2$-error on private query answers corresponds exactly to the $\ell_2$-norm estimation error. A key difference to the approach in [9] and related works is that we project the noisy answers onto the set $\mathcal{K} := \mathcal{B}_1^d(0, L\sqrt{s})$, which is a (coarse) convex relaxation of $\mathrm{conv}(\mathcal{S}_s^d)$. This is crucial to make our algorithm efficiently implementable. Due to space limitations, proofs from this section have been deferred to Appendix B.

---

**Algorithm 1** `Projection_Mechanism`$(\bar{z}(S), \varepsilon, \delta, n)$

---

**Require:** Vector $\bar{z}(S) = \frac{1}{n} \sum_{i=1}^n z_i$ from dataset $S \in (\mathcal{S}_s^d)^n$; $\varepsilon, \delta \ge 0$, privacy parameters

$\tilde{z} = \bar{z}(S) + \xi$, with $\xi \sim \begin{cases} \mathsf{Lap}(\sigma)^{\otimes d} & \text{with } \sigma = \left(\frac{2L\sqrt{s}}{n\varepsilon}\right) \text{ if } \delta = 0\,, \\ \mathcal{N}(0, \sigma^2 I) & \text{with } \sigma^2 = \frac{8L^2 \ln(1.25/\delta)}{(n\varepsilon)^2} \text{ if } \delta > 0\,. \end{cases}$

    **return** $\hat{z} = \mathrm{argmin}\{\|z - \tilde{z}\|_2 : z \in \mathcal{K}\}$, where $\mathcal{K} := \mathcal{B}_1^d(0, L\sqrt{s})$

---

**Lemma 3.1.** *In Algorithm 1, it holds that* $\|\hat{z} - \bar{z}(S)\|_2 \le \sqrt{2L\|\xi\|_\infty \sqrt{s}}$*, almost surely.*

We now provide the privacy and accuracy guarantees of Algorithm 1.

**Theorem 3.2.** *For $\delta = 0$, Algorithm 1 is $\varepsilon$-DP, and with probability $1 - \beta$:*

$$\|\hat{z} - \bar{z}(S)\|_2 \lesssim L \cdot \min\left\{ \frac{\sqrt{sd}\ln(d/\beta)}{n\varepsilon}, \sqrt{\frac{s\ln(d/\beta)}{n\varepsilon}} \right\}.$$

**Theorem 3.3.** *For $\delta > 0$, Algorithm 1 is $(\varepsilon, \delta)$-DP, and with probability $1 - \beta$:*

$$\|\hat{z} - \bar{z}(S)\|_2 \lesssim L \cdot \min\left\{ \frac{(\sqrt{d}+\sqrt{\log(1/\beta)})\sqrt{\ln(1/\delta)}}{n\varepsilon}, \frac{(s\log(1/\delta)\log(d/\beta))^{1/4}}{\sqrt{n\varepsilon}} \right\}.$$

**Sharper Upper Bound via Compressed Sensing** In Appendix B.4 we propose a faster mean estimation approximate-DP algorithm. Its rate nearly matches the lower bound we will prove in Theorem 4.4. We believe that this rate is essentially optimal. This algorithm projects the data average into a low dimensional subspace (via a random projection matrix), and uses compressed sensing to recover a noisy version of this projection: this way, noise provides privacy, which is further boosted by the random projection, and the accuracy follows from an application of the stable and noisy recovery properties of compressed sensing [10], together with the Approximate Carathéodory Theorem.

## 4 Lower Bounds for DP Mean Estimation with Sparse Data

We provide matching lower bounds to those from Section 3. Moreover, although the stated lower bounds are for mean estimation, known reductions imply analogous lower bounds for DP-ERM and DP-SCO [8, 19]. First, for pure-DP we provide a packing-type construction based on sparse vectors. This is used in a novel block-diagonal construction, which provides the right low/high-dimensional transition. On the other hand, for approximate-DP, a block diagonal reduction with existing fingerprinting codes [38, 13], suffices to obtain lower bounds that exhibit a nearly tight low/high-dimensional transition. For simplicity, we consider the case of $L = 1$, i.e., $\mathcal{S}_s^d = \{z \in \mathbb{R}^d : \|z\|_0 \le s, \|z\|_2 \le 1\}$; it is easy to see that any lower bound scales linearly in $L$. We defer proofs from this section to Appendix C.

### 4.1 Lower Bounds for Pure-DP

Our main lower bound for pure-DP mechanisms is as follows.

**Theorem 4.1.** *Let $\varepsilon > 0$ and $s < d/2$. Then the empirical mean estimation problem over $\mathcal{S}_s^d$ satisfies*

$$\inf_{\mathcal{A} : \varepsilon\text{-}DP} \sup_{S \in (\mathcal{S}_s^d)^n} \mathbb{P}\left[\|\mathcal{A}(S) - \bar{z}(S)\|_2 \gtrsim \min\left\{1, \sqrt{\tfrac{s\log(d/[\varepsilon n])}{\varepsilon n}}, \tfrac{\sqrt{sd}}{\varepsilon n}\right\}\right] \gtrsim 1.$$

The statement above—as well as those which follow—should be read as "for all DP algorithms $\mathcal{A}$, there exists a dataset $S$, such that the mean estimation error is lower bounded by $\alpha(n, d, \varepsilon, \delta)$ with probability at least $\beta(n, d, \varepsilon, \delta)$" (where in this case $\alpha \gtrsim \min\left\{1, \sqrt{\tfrac{s\log(d/[\varepsilon n])}{\varepsilon n}}, \tfrac{\sqrt{sd}}{\varepsilon n}\right\}$ and $\beta \gtrsim 1$).

We also introduce a strengthening of the worst case lower bound, based on hard distributions.

**Definition 4.2.** We say that a probability $\mu$ over $\mathcal{Z}^n$ induces an $(\alpha, \beta)$-*distributional lower bound* for $(\varepsilon, \delta)$-DP mean estimation if $\inf_{\mathcal{A} : (\varepsilon, \delta)\text{-DP}} \mathbb{P}_{S \sim \mu, \mathcal{A}}\left[\|\mathcal{A}(S) - \bar{z}(S)\|_2 \geq \alpha\right] \geq \beta$.

Note this type of lower bound readily implies a worst case lower bound. On the other hand, while the existence of hard distributions follows by the existence of hard datasets (by Yao's minimax principle), we provide explicit constructions of these distributions, for the sake of clarity.

Theorem 4.1 follows by combining the two results that we provide next. First, and our main technical innovation in the sparse case is a block-diagonal dataset bootstrapping construction, which turns a low-dimensional lower bound into a high-dimensional one.

**Lemma 4.3** (Block-Diagonal Lower Bound Bootstrapping). *Let $n_0, t \in \mathbb{N}$. Let $\mu$ be a distribution over $(\mathcal{S}_s^t)^{n_0}$ that induces an $(\alpha_0, \rho_0)$-distributional lower bound for $(\varepsilon, \delta)$-DP mean estimation. Then, for any $d \geq t$, $n \geq n_0$ and $K \leq \min\left\{\tfrac{n}{n_0}, \tfrac{d}{t}\right\}$, there exists $\tilde{\mu}$ over $(\mathcal{S}_s^d)^n$ that induces an $(\alpha, \rho)$-distributional lower bound for $(\varepsilon, \delta)$-DP mean estimation, where $\alpha \gtrsim \tfrac{\alpha_0 n_0}{n}\sqrt{\rho_0 K}$ and $\rho \geq 1 - \exp(-\rho_0/8)$.*

Note that the above result needs a base lower bound for which packing-based constructions suffice.

**Theorem 4.4.** *Let $\varepsilon > 0$ and $s < d/2$. Then there exists an $(\alpha, \rho)$-distributional lower bound for $\varepsilon$-DP mean estimation over $(\mathcal{S}_s^d)^n$ with $\alpha \gtrsim \min\left\{1, \tfrac{s\log(d/s)}{\varepsilon n}\right\}$ and $\rho = 1/2$.*

### 4.2 Lower Bounds for Approximate-DP

While the lower bound for the approximate-DP case is similarly based on the block-diagonal reduction, its base lower bound follows more directly from the dense case.

**Theorem 4.5.** *Let $\varepsilon \in (0, 1]$, $2^{-o(n)} \leq \delta \leq \tfrac{1}{n^{1+\Omega(1)}}$. Then the empirical mean estimation problem over $\mathcal{S}_s^d$ satisfies*

$$\inf_{\mathcal{A} : (\varepsilon, \delta)\text{-}DP} \sup_{S \in (\mathcal{S}_s^d)^n} \mathbb{P}\left[\|\mathcal{A}(S) - \bar{z}(S)\|_2 \gtrsim \min\left\{1, \tfrac{[s\ln(1/\delta)]^{1/4}}{\sqrt{n\varepsilon}}, \tfrac{\sqrt{d\ln(1/\delta)}}{n\varepsilon}\right\}\right] \gtrsim 1.$$

## 5 Bias Reduction Method for DP-ERM with Sparse Gradients

We now start with our study of DP-ERM with sparse gradients. We defer some proofs to Appendix E. In this section and later, we will impose subsets of the following assumptions:

(A.1) *Initial distance:* For SCO, $\|x^0 - x^*(\mathcal{D})\| \leq D$; for ERM, $\|x^0 - x^*(S)\| \leq D$.
(A.2) *Diameter bound:* $\|x - y\| \leq D$, for all $x, y \in \mathcal{X}$.
(A.3) *Convexity:* $f(\cdot, z)$ is convex, for all $z \in \mathcal{Z}$.
(A.4) *Loss range:* $f(x, z) - f(y, z) \leq B$, for all $x, y \in \mathcal{X}$, $z \in \mathcal{Z}$.
(A.5) *Lipschitzness:* $f(\cdot, z)$ is $L$-Lipschitz, for all $z \in \mathcal{Z}$.
(A.6) *Smoothness:* $\nabla f(\cdot, z)$ is $H$-Lipschitz, for all $z \in \mathcal{Z}$.
(A.7) *Individual gradient sparsity:* $\nabla f(x, z)$ is $s$-sparse, for all $x \in \mathcal{X}$ and $z \in \mathcal{Z}$.

The most natural and popular DP optimization algorithms are based on SGD. Here we show how to integrate the mean estimation algorithms from Section 3 to design a stochastic first-order oracle that can be readily used by any stochastic first-order method. The key challenge here is that estimators from Section 3 are inherently biased, which is known to dramatically deteriorate the convergence rates. Hence, we start by introducing a bias reduction method.

---

**Algorithm 2** `Subsampled_Bias-Reduced_Gradient_Estimator`$(x, S, N, \varepsilon, \delta)$

---

**Require:** Dataset $S = (z_1, \ldots, z_n) \in \mathcal{Z}^n$, $\varepsilon, \delta > 0$ privacy parameters, $L$-Lipschitz loss $f(x, z)$
with $s$-sparse gradient, $x \in \mathcal{X}$, batch size parameter $N \sim \mathsf{TGeom}(M)$ with $M = \lfloor \log_2(n) \rfloor - 1$
Let $B \sim \mathsf{Unif}\left(\binom{n}{2^{N+1}}\right)$, $O, E$ a partition of $B$ with $|O| = |E| = 2^N$, $I \sim \mathsf{Unif}([n])$
$G_{N+1}^+(x, B) = \texttt{Projection\_Mechanism}(\nabla F_B(x), \varepsilon/4, \delta/4, 2^{N+1})$ (Algorithm 1)
$G_N^-(x, O) = \texttt{Projection\_Mechanism}(\nabla F_O(x), \varepsilon/4, \delta/4, 2^N)$
$G_N^-(x, E) = \texttt{Projection\_Mechanism}(\nabla F_E(x), \varepsilon/4, \delta/4, 2^N)$
$G_0(x, I) = \texttt{Projection\_Mechanism}(\nabla f(x, z_I), \varepsilon/4, \delta/4, 1)$
**Return** (below $p_k = \mathbb{P}[\mathsf{TGeom}(M) = k]$)

$$\mathcal{G}(x) = \frac{1}{p_N}\left(G_{N+1}^+(x, B) - \frac{1}{2}\left(G_N^-(x, O) + G_N^-(x, E)\right)\right) + G_0(x, I)$$

---

**Algorithm 3** `Subsampled_Bias-Reduced_Sparse_SGD`$(x^0, S, \varepsilon, \delta)$

---

**Require:** Initialization $x^0 \in \mathcal{X}$; Dataset $S = (z_1, \ldots, z_n) \in \mathcal{Z}^n$; $\varepsilon, \delta$, privacy parameters; stepsize
$\eta > 0$; gradient oracle for $L$-Lipschitz and with $s$-sparse gradient loss $f(\cdot, z)$
$t \leftarrow -1$
**while** $\sqrt{2 \ln\left(\frac{4}{\delta}\right) \sum_{s=0}^{t-1} \left(\frac{3 \cdot 2^{N_s+1}+1}{16n}\right)^2} + \frac{\varepsilon}{2} \sum_{s=0}^{t-1} \left(\frac{3 \cdot 2^{N_s+1}+1}{16n}\right)^2 \leq \frac{1}{2}$ and $\sum_{s=0}^{t-1} \frac{3 \cdot 2^{N_s+1}+1}{16n} \leq \frac{1}{4}$
**do**
    $t \leftarrow t + 1$
    $N_t \sim \mathsf{TGeom}(M)$ where $M = \lfloor \log_2(n) \rfloor - 1$
    $\mathcal{G}(x^t) = \texttt{Subsampled\_Bias-Reduced\_Gradient\_Estimator}(x^t, S, N_t, \varepsilon/8, \delta/4)$ (Alg. 2)
    $x^{t+1} = \Pi_{\mathcal{X}}\left[x^t - \eta \mathcal{G}(x^t)\right]$
**end while**
**return** $\begin{cases} \bar{x} = \frac{1}{t+1} \sum_{s=0}^t x^s & \text{if } f(\cdot, z) \text{ is convex}, \\ x^{\hat{t}} \text{ where } \hat{t} \sim \mathsf{Unif}(\{0, \ldots, T\}) & \text{if } f(\cdot, z) \text{ is not convex}. \end{cases}$

---

### 5.1 Subsampled Bias-Reduced Gradient Estimator for DP-ERM

We propose Algorithm 2, inspired by a debiasing technique proposed in [14]. The idea is the following: we know that the projection mechanism[2] would provide more accurate gradient estimators with larger sample sizes, and we will see that its bias improves analogously. We choose our batch size as a random variable with exponentially increasing range, and given such a realization we subtract the projection mechanism applied to the whole batch minus the same mechanism applied to both halves of this batch.[3] This subtraction, together with a multiplicative and additive correction, results in the expected value of the outcome $\mathcal{G}(x)$ corresponding to the estimator with the largest batch size, leading to its expected accuracy being boosted by such large sample size, without necessarily utilizing such amount of data (in fact, the probability of such batch size being picked is polynomially smaller, compared to the smallest possible one). The caveat with this technique, as we will see, relates to a heavy-tailed distribution of outcomes, and therefore great care is needed for its analysis.

Instrumental to our analysis is the following *truncated geometric distribution* with parameter $M \in \mathbb{N}$, whose law will be denoted by $\mathsf{TGeom}(M)$: we say $N \sim \mathsf{TGeom}(M)$ if it is supported on $\{0, \ldots, M\}$, and takes value $k$ with probability $p_k := C_M/2^k$, where $C_M = (2(1 - 2^{-(M+1)}))^{-1}$, is the normalizing constant. Note that $1/2 \leq C_M \leq 1$, thus it is bounded away from 0 and $+\infty$.

We propose Algorithm 3, which interacts with the oracle given in Algorithm 2. For convenience, we will denote the random realization from the truncated geometric distribution used in iteration $t$ by $N_t$. The idea is that, using the fully adaptive composition property of DP [15], we can run the method until our privacy budget is exhausted. Due to technical reasons, related to the bias reduction, we need

---

[2]Note that we use the projection mechanism (Algorithm 1) as subroutine for Algorithm 2 only to have a self-contained presentation in the main body of the paper. We will analyze and state the sharper bounds obtained with Algorithm 5 as subroutine.

[3]We follow the Blanchet-Glynn notation of $O$ and $E$ to denote the 'odd' and 'even' terms for the batch partition [14]; this partitioning is arbitrary.

to shift by one the termination condition in the algorithm. In particular, our algorithm goes over the reduced privacy budget of $(\varepsilon/2, \delta/2)$. The additional slack in the privacy budget guarantees that even with the extra oracle call the algorithm respects the privacy constraint.

**Lemma 5.1.** *Algorithm 3 is $(\varepsilon, \delta)$-DP.*

## 5.2 Bias and Moment Estimates for the Debiased Gradient Estimator

We provide bias and second moment estimates for our debiased estimator of the empirical gradient. In summary, we show that this estimator has bias matching that of the full-batch gradient estimator, while at the same time its second moment is bounded by a mild function of the problem parameters.

**Lemma 5.2.** *Let $d \gtrsim \frac{n\varepsilon\sqrt{s\ln(d/s)}}{\sqrt{\ln(1/\delta)}}$. Algorithm 2, enjoys bias and second moment bounds*

$$\left\|\mathbb{E}[\mathcal{G}(x) - \nabla F_S(x)|x]\right\| \lesssim \frac{L[s\ln(d/s)\ln(1/\delta)]^{1/4}}{\sqrt{n\varepsilon}} =: b,$$

$$\mathbb{E}[\|\mathcal{G}(x)\|^2|x] \lesssim \frac{L^2\ln(n)\sqrt{s\ln(d/s)\ln(1/\delta)}}{\varepsilon} =: \nu^2.$$

*Proof.* For simplicity, we assume without loss of generality that $n$ is a power of 2, so that $2^{M+1} = n$.

**Bias.** Let, for $k = 0, \ldots, M$, $G_{k+1}^+(x) = \mathbb{E}[G_{N+1}^+(x, B) \mid N = k, x]$, and

$$G_k^-(x) = \mathbb{E}[G_N^-(x, E) \mid N = k, x] = \mathbb{E}[G_N^-(x, O) \mid N = k, x],$$

where the last equality follows from the identical distribution of $O$ and $E$. Noting further that $G_k^+(x) = G_k^-(x)$ (which follows from the uniform sampling and the cardinality of the used datapoints), and using the law of total probability, we have

$$
\begin{aligned}
\mathbb{E}[\mathcal{G}(x) \mid x] &= \textstyle\sum_{k=0}^M \left(G_k^+(x) - G_{k-1}^-(x)\right) + \mathbb{E}[G_0(x, I) \mid x] \\
&= G_{M+1}^+(x) - G_0^-(x) + \mathbb{E}[G_0(x, I) \mid x] \\
&= \mathbb{E}[G_{M+1}^+(x) - \nabla F_S(x)|x] + \nabla F_S(x),
\end{aligned}
$$

where we also used that $\mathbb{E}[G_0(x, I) \mid x] = G_0^-(x)$ (since $I$ is a singleton). Next, by Theorem B.1

$$\|\mathbb{E}[\mathcal{G}(x) \mid x] - \nabla F_S(x)\| \leq \|\mathbb{E}[G_{M+1}^+(x) - \nabla F_S(x)|x]\| \lesssim L\frac{[s\ln(d/s)\ln(1/\delta)]^{1/4}}{\sqrt{n\varepsilon}}.$$

**Second moment bound.** Using the law of total probability, and that $O, E$ are a partition of $B$:

$$
\mathbb{E}[\|\mathcal{G}(x)\|^2 \mid x] = \sum_{k=0}^M p_k \mathbb{E}\bigg[\bigg\|\frac{1}{p_k}[G_{N+1}^+(x, B) - \nabla F_B(x)]
$$

$$
- \frac{1}{2p_k}\big[G_N^-(x, O) - \nabla F_O(x) + G_N^-(x, E) - \nabla F_E(x)\big] + G_0(x, I)\bigg\|^2 \bigg|x, N = k\bigg]
$$

$$
\leq 2\mathbb{E}[\|G_0(x, I)\|^2 \mid x] + 4\sum_{k=0}^M \frac{1}{p_k}\mathbb{E}\bigg[\bigg\|G_{N+1}^+(x, B) - \nabla F_B(x)\bigg\|^2\bigg|x, N = k\bigg]
$$

$$
+ \sum_{k=0}^M \frac{1}{p_k}\mathbb{E}\bigg[\bigg\|G_N^-(x, O) - \nabla F_O(x)\bigg\|^2 + \bigg\|G_N^-(x, E) - \nabla F_E(x)\bigg\|^2\bigg|x, N = k\bigg].
$$

We now use Theorem B.1, to conclude that

$$\mathbb{E}\bigg[\bigg\|G_{N+1}^+(x, B) - \nabla F_B(x)\bigg\|^2 \bigg| x, N = k\bigg] \lesssim \frac{L^2\sqrt{s\ln(d/s)\ln(1/\delta)}}{2^{k+1}\varepsilon}$$

$$\max_{A\in\{O,E\}}\left\{\mathbb{E}\bigg[\bigg\|G_N^-(x, A) - \nabla F_A(x)\bigg\|^2 \bigg| x, N = k\bigg]\right\} \lesssim \frac{L^2\sqrt{s\ln(d/s)\ln(1/\delta)}}{2^k\varepsilon}$$

$$\mathbb{E}\bigg[\bigg\|G_0(x, I)\bigg\|^2 \bigg| x\bigg] \lesssim \frac{L^2\sqrt{s\ln(d/s)\ln(1/\delta)}}{\varepsilon}.$$

Recalling that $M + 1 = \log_2 n$ and $p_k = 2^{-k}$, these bounds readily imply that $\mathbb{E}\|\mathcal{G}(x)\|^2 \lesssim \nu^2$. $\quad\square$

## 5.3 Accuracy Guarantees for Subsampled Bias-Reduced Sparse SGD

The previous results provide useful information about the privacy, bias, and second-moment of our proposed oracle. Our goal now is to provide excess risk rates for DP-ERM. For this, we need to prove the algorithm runs for long enough, i.e., a lower bound on the stopping time of Algorithm 3,

$$T := \inf\left\{ t : \frac{\varepsilon}{2} < \varepsilon\Big(2\ln\big(\tfrac{4}{\delta}\big)\sum_{s=0}^{t}\big(\tfrac{3\cdot 2^{N_s+1}+1}{16n}\big)^2\Big)^{1/2} + \frac{\varepsilon^2}{2}\sum_{s=0}^{t}\frac{3\cdot 2^{N_s+1}+1}{16n} \ \ \text{or} \ \ \frac{\delta}{4} < \sum_{s=0}^{t}\frac{(3\cdot 2^{N_s+1}+1)\delta}{16n} \right\}. \tag{2}$$

The proof of Theorem 5.2 implies that moments of $\mathcal{G}$ increase exponentially in $M$. This heavy-tailed behavior implies that $T$ may not concentrate strongly enough to obtain high probability lower bounds for $T$. What we will do instead is showing that *with constant probability $T$* behaves as desired.

To justify the approach, let us provide a simple in-expectation bound on how the privacy budget accumulates in the definition of $T$: letting $\varepsilon_t = (3 \cdot 2^{N_t+1} + 1)\varepsilon/[16n]$, we have that

$$\mathbb{E}\Big[\sum_{s=0}^{t}\varepsilon_s^2\Big] = \frac{(t+1)\varepsilon^2}{(16n)^2}\mathbb{E}\Big[(3\cdot 2^{N_1+1}+1)^2\Big] \leq \frac{2(t+1)\varepsilon^2}{(16n)^2}\Big(9\mathbb{E}[2^{2(N_1+1)}]+1\Big) \lesssim \frac{t\varepsilon^2}{n},$$

where in the last step we used that $\mathbb{E}\big[2^{2(N_1+1)}\big] = C_M \sum_{k=1}^{M+1} 2^k \lesssim n$. This in-expectation analysis can be used in combination with ideas from stopping times to establish bounds for $T$.

**Lemma 5.3.** *Let $0 < \delta < 1/n^2$. Let $T$ be the stopping time defined in eqn. (2). Then, there exists $t = Cn/\log(2/\delta)$ (with $C > 0$ an absolute constant) such that $\mathbb{P}[T \leq t] \leq 1/4$. On the other hand,*

$$\frac{n^2}{(n+1)\ln(4/\delta)} - 1 \leq \mathbb{E}[T] \leq \frac{64n}{9\ln(4/\delta)}.$$

With our bounds on $T$, further analysis involving regret bounds on randomly stopped SGD yields the following bounds for convex and nonconvex losses. See Theorem E.2 and Theorem E.3 for details.

**Theorem 5.4.** *Consider a* (SO) *problem under initial distance (Item (A.1)), Lipschitzness (Item (A.5)) and gradient sparsity (Item (A.7)) assumptions.*

- *In the convex case (Item (A.3)), Algorithm 3 satisfies*

$$\mathbb{P}\Big[F_S(\hat{x}) - F_S(x^*(S)) \lesssim LD\frac{\sqrt{\ln n}[s\ln(d/s)\ln^3(1/\delta)]^{1/4}}{\sqrt{\varepsilon n}}\Big] \geq \frac{1}{2}.$$

- *In the nonconvex case, additionally assuming smoothness (Item (A.6)) and the following* initial suboptimality assumption*: namely, that given our initialization $x^0 \in \mathbb{R}^d$, there exists $\Gamma > 0$ such that $F_S(x^0) - F_S(x^*(S)) \leq \Gamma$; Algorithm 3 satisfies*

$$\mathbb{P}\Big[\|\nabla F_S(x^{\hat{t}})\|_2^2 \lesssim \big(\sqrt{\Gamma H}L\sqrt{\ln(n)\ln(1/\delta)} + L^2\big)\frac{[s\ln(d/s)\ln(1/\delta)]^{1/4}}{\sqrt{\varepsilon n}}\Big] \geq \frac{1}{2}.$$

**Boosting the Confidence of the Bias-Reduced SGD**   To conclude, in Appendix F we provide a boosting algorithm that can exponentially amplify the success probability of Algorithm 3. The approach is based on making parallel runs of the method and using private model selection to obtain the best performing model.

# 6 DP Convex Optimization with Sparse Gradients via Regularized Output Perturbation

We conclude our work introducing another class of algorithms that attains nearly optimal rates for approximate-DP ERM and SO in the convex setting. These algorithms are based on solving a regularized ERM problem and privatizing its output by an output perturbation method. The main innovation of this technique is that we reduce the noise error by a $\|\cdot\|_\infty$-projection. This type of projection leverages the concentration of the noise in high-dimensions. We carry out an analysis that also leverages the convexity of the risk and the gradient sparsity to obtain these rates. The full description is included in Algorithm 4. We defer missing proofs from this section, as well as additional results, to Appendix G.

---
**Algorithm 4** `Output_Perturbation`

---

**Require:** Dataset $S = (z_1, \ldots, z_n) \in \mathcal{Z}^n$, $\varepsilon, \delta \geq 0$ privacy params., $f(\cdot, z)$ $L$-Lipschitz convex function (if $\delta = 0$ further assume $H$-smooth) with $s$-sparse gradient, $\lambda \geq 0$ regularization param.

Let $x_\lambda^*(S) = \operatorname{argmin}_{x \in \mathcal{X}} F_S^\lambda(x)$, where $F_S^\lambda(x) := \left[ F_S(x) + \frac{\lambda}{2} \|x\|_2^2 \right]$

$\tilde{x} = x_\lambda^*(S) + \xi$, with $\xi \sim \begin{cases} \mathsf{Lap}(\sigma)^{\otimes d} & \text{with } \sigma = \frac{2\sqrt{2s}L}{\lambda \varepsilon n}\left(\frac{2H}{\lambda} + 1\right) \text{ if } \delta = 0\,, \\ \mathcal{N}(0, \sigma^2 I) & \text{with } \sigma^2 = \frac{8L^2 \ln(1.25/\delta)}{[\lambda \varepsilon n]^2} \text{ if } \delta > 0\,. \end{cases}$

**return** $\hat{x} = \operatorname{argmin}_{x \in \mathcal{X}} \|x - \tilde{x}\|_\infty$ (breaking ties arbitrarily)

---

**Theorem 6.1.** *Consider an ERM problem under assumptions: initial distance (Item (A.1)), convexity (Item (A.3)), Lipchitzness (Item (A.5)) and gradient sparsity (Item (A.7)). Then, Algorithm 4 is $(\varepsilon, \delta)$-DP, and it satisfies the following excess risk guarantees, for any $0 < \beta < 1$:*

- *If $\delta = 0$, and under the additional assumption of smoothness (A.6) and unconstrained domain, $\mathcal{X} = \mathbb{R}^d$, then selecting $\lambda = \left( \frac{L^2 H}{D^2} \frac{s \log(d/\beta)}{\varepsilon n} \right)^{1/3}$, it holds with probability $1 - \beta$ that*

$$F_S(\hat{x}) - F_S(x^*(S)) \lesssim L^{2/3} H^{1/3} D^{4/3} \left( \frac{s \log(d/\beta)}{\varepsilon n} \right)^{1/3}.$$

- *If $\delta > 0$ then selecting $\lambda = \frac{L}{D} \cdot \frac{[s \log(1/\delta) \log(d/\beta)]^{1/4}}{\sqrt{\varepsilon n}}$, we have with probability $1 - \beta$ that*

$$F_S(\hat{x}) - F_S(x^*(S)) \lesssim LD \cdot \frac{(s \log(1/\delta) \log(d/\beta))^{1/4}}{\sqrt{\varepsilon n}}.$$

**Remark 6.2.** *For approximate-DP, the theorem above can also be proved if we replace assumption (Item (A.1)) by the diameter assumption (Item (A.2)). On the other hand, for the pure-DP case it is a natural question whether the smoothness assumption is essential. In Appendix G.3, we provide a version of the exponential mechanism that works without the smoothness and unconstrained domain assumptions. This algorithm is inefficient and it does require an structural assumption on the feasible set, but it illustrates the possibilities of more general results in the pure-DP setting.*

We note that the proposed output perturbation approach (Algorithm 4) leads to nearly optimal population risk bounds for approximate-DP, by a different tuning of the regularization parameter $\lambda$.

**Theorem 6.3.** *Consider a problem (SO) under bounded initial distance (Item (A.1)) (or bounded diameter, Item (A.2), if $\delta > 0$), convexity (Item (A.3)), Lipschitzness (Item (A.5)), bounded range (Item (A.4)), and gradient sparsity (Item (A.7)). Then, Algorithm 4 is $(\varepsilon, \delta)$-DP, and for $0 < \beta < 1$,*

- *If $\delta = 0$, and under the additional assumption of smoothness (A.6) and unconstrained domain, $\mathcal{X} = \mathbb{R}^d$. Selecting $\lambda = \left( \frac{L^2 H}{D^2} \frac{s \log(d/\beta)}{\varepsilon n} \right)^{1/3}$, then with probability $1 - \beta$*

$$F_S(\hat{x}) - F_S(x^*(\mathcal{D})) \lesssim L^{2/3} H^{1/3} D^{4/3} \left( \frac{s \log(d/\beta)}{\varepsilon n} \right)^{1/3} + B\sqrt{\frac{\ln(1/\beta)}{n}}.$$

- *If $\delta > 0$. Selecting $\lambda = \frac{L}{D} \left( \frac{\ln(n) \ln(1/\beta)}{n} + \frac{\sqrt{s \ln(1/\delta) \ln(d/\beta)}}{\varepsilon n} \right)^{1/2}$, then with probability $1 - \beta$*

$$F_\mathcal{D}(\hat{x}) - F_\mathcal{D}(x^*(\mathcal{D})) \lesssim LD \frac{[s \ln(1/\delta) \log(d/\beta)]^{1/4}}{\sqrt{\varepsilon n}} + (LD\sqrt{\ln n} + B)\sqrt{\frac{\ln(1/\beta)}{n}}.$$

## Acknowledgments and Disclosure of Funding

C.G.'s research was partially supported by INRIA Associate Teams project, ANID FONDECYT 1210362 grant, ANID Anillo ACT210005 grant, and National Center for Artificial Intelligence CENIA FB210017, Basal ANID.

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

# Appendix

## A   Auxiliary Privacy Results

The privacy and accuracy of some of the perturbation based methods we use to privatize our algorithms are based on the following simple facts (see, e.g., [1]).

**Fact A.1** (Laplace & Gaussian mechanisms). *For all $g : \mathcal{Z}^n \mapsto \mathbb{R}^d$*

(a) *If the $\ell_1$-sensitivity of $g$ is bounded, i.e., $\Delta_1^g := \sup_{S \simeq S'} \|g(S) - g(S')\|_1 < +\infty$, then $\mathcal{A}_{\mathsf{Lap}}^g(S) := g(S) + \xi$ where $\xi \sim \mathsf{Lap}^{\otimes d}(\Delta_1^g/\varepsilon)$ is $\varepsilon$-DP.*

(b) *If the $\ell_2$-sensitivity of $g$ is bounded, i.e., $\Delta_2^g := \sup_{S \simeq S'} \|g(S) - g(S')\|_2 < +\infty$, then $\mathcal{A}_{\mathcal{N}}^g(S) := g(S) + \xi$, where $\xi \sim \mathcal{N}(0, \sigma^2 I)$ for $\sigma \geq \frac{\Delta_2^g \sqrt{2 \log(1.25/\delta)}}{\varepsilon}$ is $(\varepsilon, \delta)$-DP.*

**Fact A.2** (Laplace & Gaussian concentration). *Let $\sigma > 0$ and $0 < \beta < 1$.*

(a) *For $\xi \sim \mathsf{Lap}(\sigma)^{\otimes d}$: (i) $\|\xi\|_\infty \lesssim \sigma \log(d/\beta)$ holds with probability $1 - \beta$, and (ii) $\|\xi\|_2 \lesssim \sigma \sqrt{d} \log(d/\beta)$ holds with probability $1 - \beta$.*

(b) *For $\xi \sim \mathcal{N}(0, \sigma^2 I)$, (i) $\|\xi\|_\infty \lesssim \sigma \sqrt{\log(d/\beta)}$ holds with probability $1 - \beta$, (ii) $\|\xi\|_2 \lesssim \sigma(\sqrt{d} + \sqrt{\log(1/\beta)})$ holds with probability $1 - \beta$, and (iii) $\mathbb{E}\|\xi\|_2^2 = d\sigma^2$.*

We note the existence of packing sets of sparse vectors (e.g., [39, 40]). Denote by $\mathcal{C}_s^d$ the set of all $s$-sparse vectors in $\{0, 1/\sqrt{s}\}^d$; note that $\mathcal{C}_s^d \subseteq \mathcal{S}_s^d$.

**Lemma A.3.** *For all $s$ and $d$ such that $s \leq d/2$, there exists a subset $\mathcal{P} \subseteq \mathcal{C}_s^d$ such that $|\mathcal{P}| \geq (d/s - 1/2)^{s/2}$ and for all $u, v \in \mathcal{P}$, it holds that $\|u - v\|_2 \geq 1/\sqrt{2}$.*

*Proof.* This follows from a simple packing-based construction (see, e.g., [40]). There are $\binom{d}{s}$ vectors in $\mathcal{C}_s^d$, and for each vector $v \in \mathcal{C}_s^d$, there are at most $\binom{d}{\lfloor s/2 \rfloor}$ many vectors $u \in \mathcal{C}_s^d$ such that $\|u - v\|_0 \leq s/2$ and hence $\|u - v\|_2 \leq 1/\sqrt{2}$. Thus, we can greedily pick vectors to be $C$, guaranteeing that all vectors $u, v \in \mathcal{C}_s^d$ satisfy $\|u - v\|_0 > s/2$, and have $|C| \geq \binom{d}{s}/\binom{d}{\lfloor s/2 \rfloor} \geq \left(\frac{d}{s} - \frac{1}{2}\right)^{s/2}$. $\qquad\square$

For completeness, we provide a classical dataset bootstrapping argument used for DP mean estimation lower bounds [8]. Whereas in the original reference this bootstrapping is achieved by appending dummy vectors which mutually cancel out with the goal of maintaining the structure of vectors, we simply append zero vectors as dummies as we do not need to satisfy an exact sparsity pattern.

**Lemma A.4** (Dataset bootstrapping argument from [8]). *Suppose for some $n$, there exists a mechanism $\mathcal{A}$ such that for all $S \in (\mathcal{S}_s^d)^n$, it holds with probability at least $1/2$ that $\|\mathcal{A}(S) - \bar{z}(S)\|_2 \leq C$, for some $C \geq 0$. Then for all $n^* < n$, there exists a mechanism $\mathcal{A}'$ such that for all $S' \in (\mathcal{S}_s^d)^{n^*}$, it holds with probability at least $1/2$ that $\|\mathcal{A}(S') - \bar{z}(S')\|_2 \leq C\frac{n}{n^*}$. Furthermore, $\mathcal{A}'$ satisfies the same privacy guarantees as $\mathcal{A}$, namely if $\mathcal{A}$ is $\varepsilon$-DP (or $(\varepsilon, \delta)$-DP), then so is $\mathcal{A}'$.*

*Proof.* Given mechanism $\mathcal{A}$, consider mechanism $\mathcal{A}'$ that for any dataset $S' \in (\mathcal{S}_s^d)^{n^*}$, builds dataset $S$ by adding $n - n^*$ copies of $\mathbf{0}$ to $S'$ and returns $\frac{n}{n^*}\mathcal{A}(S)$. From the guarantees of $\mathcal{A}$, it holds that $\mathbb{P}[\|\mathcal{A}(S) - \bar{z}(S)\|_2 \leq C] \geq \frac{1}{2}$. Since $\mathcal{A}'(S') = \frac{n}{n^*}\mathcal{A}(S)$ and $\bar{z}(S') = \frac{n}{n^*}\bar{z}(S)$, it follows that

$$\mathbb{P}\left[\|\mathcal{A}'(S') - \bar{z}(S')\|_2 \leq C\frac{n}{n^*}\right] \geq \frac{1}{2}.$$

Since $\mathcal{A}'$ just applies $\mathcal{A}$ once, it follows that $\mathcal{A}'$ satisfies the same privacy guarantee as $\mathcal{A}$. $\qquad\square$

Next we provide a generic reduction of existence of packing sets with pure-DP mean estimation lower bounds. Note however that the lower bounds we state work on the distributional sense.

**Lemma A.5** (Packing-based mean estimation lower bound, adapted from [12, 8]). *Let $\mathcal{P} \subseteq \mathbb{R}^d$ be an $\alpha_0$-packing set of vectors with $|\mathcal{P}| = p$. Then, there exists a distribution $\mu$ over $\mathcal{P}^n$ that induces an $(\alpha, \rho)$-distributional lower bound for $\varepsilon$-DP mean estimation with $\alpha = \frac{\alpha_0}{2} \min \left\{ 1, \frac{\log(p/2)}{\varepsilon n} \right\}$ and $\rho = 1/2$.*

*Proof.* Let $n^* = \frac{\log(p/2)}{\varepsilon}$. First, consider the case where $n < n^*$. We construct $p$ datasets $S_1, \ldots S_p$ where $S_l$ consists of $n$ copies of $z_l$, and define $\mu = \text{Unif}(\{S_1, \ldots, S_p\})$. Note that for all $k \neq l$, it holds that $\|\bar{z}(S_k) - \bar{z}(S_l)\|_2 \geq \alpha_0$. Suppose $\mu$ does not induce a distributional lower bound. Then there exists $\mathcal{A}$ which is $\varepsilon$-DP and has $\ell_2$-accuracy better than $\alpha_0/2$ w.p. at least $1/2$: this implies in particular that

$$\mathbb{P}_{l \sim \text{Unif}([p])} \left[ \mathcal{A}(S_l) \in \mathcal{B}_2^d(z_l), \frac{\alpha_0}{2}) \right] \geq \frac{1}{2}.$$

For all distinct $k$, $l$, the datasets $S_k$ and $S_l$ differ in all $n$ entries, and hence for any $\varepsilon$-DP mechanism $\mathcal{A}$, it holds that $\mathbb{P}[\mathcal{A}(S_l) \in \mathcal{B}_2^d(z_k, \frac{\alpha_0}{2})] \geq \frac{1}{2} e^{-\varepsilon n}$. However, by construction, $\mathcal{B}_2^d(z_l, \frac{\alpha_0}{2})$ are pairwise disjoint. Hence,

$$1 \geq \sum_{k=1}^{p} \mathbb{P}_{S \sim \mu}[\mathcal{A}(S) \in \mathcal{B}_2^d(z_k, \alpha_0/2)] = \sum_{j=1}^{p} \sum_{k=1}^{p} \mathbb{P}_{S \sim \mu}[\mathcal{A}(S) \in \mathcal{B}_2^d(z_k, \alpha_0/2)|S = z_j] \frac{1}{p}$$

$$\geq \frac{e^{-\varepsilon n}}{p} \sum_{j=1}^{p} \sum_{k=1}^{p} \mathbb{P}_{S \sim \mu}[\mathcal{A}(S) \in \mathcal{B}_2^d(z_k, \alpha_0/2)|S = z_k] \geq \frac{e^{-\varepsilon n} p}{2}.$$

Thus, we get that $n \geq \frac{\log(p/2)}{\varepsilon}$, which is a contradiction since we assumed $n < n^*$. Hence, $\mu$ induces an $(\alpha_0/2, 1/2)$-distributional lower bound for $\varepsilon$-DP mean estimation.

Next, consider the case where $n > n^*$. Then the previous argument together with Lemma A.4 implies an $(\alpha, \rho)$-lower bounded, where $\alpha = \frac{n^*}{2n}$ and $\rho = 1/2$, as desired. $\qquad\square$

We will make use of the following *fully adaptive composition* property of DP, which informally states that for a prescribed privacy budget, a composition of (adaptively chosen) mechanisms whose privacy parameters are predictable, if we stop the algorithm before the (predictable) privacy budget is exhausted, the result of the full transcript is DP.

**Theorem A.6** $((\varepsilon, \delta)$-DP Filter, [15]). *Suppose $(\mathcal{A}_t)_{t \geq 0}$ is a sequence of algorithms such that, for any $t \geq 0$, $\mathcal{A}_t$ is $(\varepsilon_t, \delta_t)$-DP, conditionally on $(\mathcal{A}_{0:t-1})$ (in particular, $(\varepsilon_t, \delta_t)_t$ is $(\mathcal{A}_t)_t$-predictable). Let $\varepsilon > 0$ and $\delta = \delta' + \delta''$ be the target privacy parameters such that $\delta' > 0, \delta'' \geq 0$. Let*

$$\varepsilon_{[0:t]} := \sqrt{2 \ln \left( \frac{1}{\delta'} \right) \sum_{s=0}^{t} \varepsilon_s^2 + \frac{1}{2} \sum_{s=0}^{t} \varepsilon_s^2}, \quad and \quad \delta_{[0:t]} := \sum_{s=0}^{t} \delta_s,$$

*and define the stopping time*

$$T((\varepsilon_t, \delta_t)_t) := \inf \left\{ t : \varepsilon < \varepsilon_{[0:t+1]} \right\} \wedge \inf \left\{ t : \delta'' < \delta_{[0:t+1]} \right\}.$$

*Then, the algorithm $\mathcal{A}_{0:T(\cdot)}(\cdot)$ is $(\varepsilon, \delta)$-DP, where $T(x) = T\big((\varepsilon_t(x), \delta_t(x))_{t \geq 0}\big)$.*

# B  Missing Proofs from Section 3

## B.1  Proof of Lemma 3.1

*Proof.* From the properties of the Euclidean projection, we have

$$\langle \hat{z} - \bar{z}(S), \hat{z} - \tilde{z} \rangle \leq 0. \tag{3}$$

Hence,

$$\|\hat{z} - \bar{z}(S)\|_2^2 = \langle \hat{z} - \bar{z}(S), \hat{z} - \tilde{z} \rangle + \langle \hat{z} - \bar{z}(S), \xi \rangle \overset{(3)}{\leq} \langle \hat{z} - \bar{z}(S), \xi \rangle$$

$$\leq 2 \cdot \max_{u \in \mathcal{K}} \langle u, \xi \rangle \leq 2 \cdot \max_{u \in \mathcal{K}} \|u\|_1 \cdot \|\xi\|_\infty = 2L \|\xi\|_\infty \sqrt{s},$$

where we used the fact that $\text{conv}(\mathcal{S}_s^d) \subseteq \mathcal{K}$. $\qquad\square$

## B.2 Proof of Theorem 3.2

*Proof.* First, the privacy follows from the $\ell_1$-sensitivity bound of the empirical mean

$$\Delta_1 = \sup_{S \simeq S'} \|\bar{z}(S) - \bar{z}(S')\|_1 = \frac{1}{n} \sup_{z, z' \in \mathcal{S}_s^d} \|z - z'\|_1 \leq \frac{2L\sqrt{s}}{n},$$

together with Theorem A.1(a).

For the accuracy, the first term follows from Theorem A.2(a)-(ii), and the fact that Euclidean projection does not increase the $\ell_2$-estimation error, and the second term follows from Lemma 3.1 with the fact that $\|\xi\|_\infty \leq O\left(\frac{L\sqrt{s}}{n\varepsilon} \cdot \log(d/\beta)\right)$ holds with probability at least $1 - \beta$, by Theorem A.2(a)-(i). $\square$

## B.3 Proof of Theorem 3.3

*Proof.* The privacy guarantee follows from the $\ell_2$-sensitivity bound of the empirical mean, $\Delta_2 = \frac{2L}{n}$, together with Theorem A.1(b). For the accuracy, the first term in the minimum follows from Theorem A.2(b)-(ii), and the fact that Euclidean projection does not increase the $\ell_2$-estimation error. The second term follows from Lemma 3.1 and Theorem A.2(b)-(ii). $\square$

## B.4 Sharper DP Mean Estimation Upper Bounds via Compressed Sensing

We propose Algorithm 5, a more accurate method for approximate-DP mean estimation based on compressed sensing [10]. The precise improvements relate to reducing the $\log(d)$ factor to $\log(d/s)$, and a faster rate dependence on the confidence $\beta$. The idea is that for sufficiently high dimensions, a small number of random measurements suffices to estimate a *noisy and approximately sparse signal*. These properties follow from existing results in compressed sensing, which provide guarantees based on the $\ell_2$-norm of the noise, and the best sparse approximation in the $\ell_2$-norm (known as $\ell_2$-$\ell_2$-stable and noisy recovery) [10]. We will exploit such robustness in two ways: regarding the noise robustness, this property is used in order to perturb our measurements, which will certify the privacy; on the other hand, the approximate recovery property is used to find a sparser approximation of our empirical mean. As the approximation is only used for analysis, we can appeal to the Approximate Caratheodory Theorem to certify the existence of a sparse vector whose sparsity increases more moderately with $n$ than the empirical average [11].

An interesting feature of this algorithm is that $\ell_1$-minimization promotes sparse solutions, and thus we expect our output to be approximately sparse: this is not a feature that we particularly exploit, but it may be relevant for computational and memory considerations. Furthermore, note that the $\ell_1$-minimization problem does not require exact optimality for the privacy guarantee, hence approximate solvers can be used without compromising privacy.

---

**Algorithm 5** Gaussian $\ell_1$-Recovery$(\bar{z}(S), \varepsilon, \delta, n)$

---

**Require:** $\bar{z}(S) = \frac{1}{n} \sum_{i \in [n]} z_i \in \mathbb{R}^d$ from dataset $S \in (\mathcal{S}_{s,d})^n$; privacy parameters $\varepsilon, \delta > 0$

$m \approx n\varepsilon \sqrt{\frac{s \ln(d/s)}{\ln(1/\delta)}}$

**return** $\hat{z} = \begin{cases} \bar{z}(S) + \xi, \text{ where } \xi \sim \mathcal{N}(0, \sigma^2 I_{d \times d}) \text{ and } \sigma^2 = \frac{8L^2 \ln(1.25/\delta)}{(n\varepsilon)^2}, & \text{if } d < m \ln^2 m, \\ \tilde{z} \cdot \mathbb{1}\{\|\tilde{z}\|_2 \leq 2L\}, \text{ where } \tilde{z} = \arg\min\{\|z\|_1 : Az = b\}, A \sim (\mathcal{N}(0, \frac{1}{m}))^{m \times d}, \\ \quad b = A\bar{z}(S) + \xi \text{ and } \xi \sim \mathcal{N}(0, \sigma^2 I_{m \times m}) \text{ with } \sigma^2 = \frac{18L^2 \ln(2.5/\delta)}{(n\varepsilon)^2}, & \text{otherwise} \end{cases}$

---

**Theorem B.1.** *If* $6 \exp\{-cm\} \leq \delta < \frac{s \ln(d/s)}{m^2}$ *(where* $c > 0$ *is a constant) and* $0 < \varepsilon \leq 1$*, then Algorithm 5 is* $(\varepsilon, \delta)$*-DP, and with probability* $1 - \delta/2 - \beta$*,*

$$\|\hat{z} - \bar{z}(S)\|_2 \lesssim L \min \left\{ \frac{(\sqrt{d} + \sqrt{\ln(1/\beta)})\sqrt{\ln(1/\delta)}}{n\varepsilon}, \frac{(s \ln(d/s) \ln(1/\delta))^{1/4}}{\sqrt{n\varepsilon}} + \frac{\sqrt{\ln(1/\beta) \ln(1/\delta)}}{n\varepsilon} \right\}. \quad (4)$$

*Moreover, we have the following second moment estimate,*

$$\mathbb{E}[\|\hat{z} - \bar{z}\|_2^2] \lesssim L^2 \min \left\{ \frac{d \ln(1/\delta)}{(n\varepsilon)^2}, \frac{\sqrt{s \ln(d/s) \ln(1/\delta)}}{n\varepsilon} \right\}.$$

*Proof.* First, if $d < m \ln^2 m$, then Algorithm 5 is $(\varepsilon, \delta)$-DP by privacy of Gaussian noise addition and the post-processing property of DP. Moreover, its (high probability and second moment) accuracy guarantees follow from Theorem A.2.

Next, if $d \geq m \ln^2 m$, we start with the privacy analysis. Let $S \simeq S'$ and suppose they only differ in their $i$th entry. We note that due to our choice of $m$, $A$ is an approximate restricted isometry with probability $1 - 3\exp\{-cm\}$ [41] (where $c$ is the same as in the theorem statement); in particular, letting $K \asymp \frac{n\varepsilon}{\sqrt{s \ln(d/s) \ln(1/\delta)}}$, we have that for all $v \in \mathbb{R}^d$ which is $(sK)$-sparse

$$\frac{1}{2}\|v\|_2 \leq \|Av\|_2 \leq \frac{3}{2}\|v\|_2.$$

Hence, due to our assumption on $\delta$, the event above has probability at least $1 - \delta/2$, and therefore

$$\|A(\bar{z} - \bar{z}')\|_2 = \frac{1}{n}\|A(z_i - z_i')\|_2 \leq \frac{3L}{n},$$

where we used the fact that $z_i - z_i'$ is $(2s)$-sparse. We conclude by the choice of $\sigma^2$ that $A\bar{z} + \xi$ is $(\varepsilon, \delta)$-DP, and thus $\tilde{z}$ is $(\varepsilon, \delta)$-DP by postprocessing.

We now proceed to the accuracy guarantee. By [10, Theorem 3.6 (b)], under the same event as stated above (which has probability $1 - \delta/2$) we have

$$\|\hat{z} - \bar{z}\|_2 \lesssim \|\xi\|_2 + \inf_{z:\, \|z\|_0 \leq sK} \|z - \bar{z}\|_2.$$

For the first term, we use Gaussian norm concentration to guarantee that with probability $1 - \beta$,

$$\|\xi\|_2 \lesssim \left(\sqrt{m} + \sqrt{\ln(1/\beta)}\right)\sigma \lesssim \left(\sqrt{Ks\ln(d/s)} + \sqrt{\ln\left(\frac{1}{\beta}\right)}\right)\frac{L\sqrt{\ln(1/\delta)}}{n\varepsilon}.$$

For the second term, by the Approximate Carátheodory Theorem [11], the infimum above is upper bounded by $O(L/\sqrt{K})$; for this, note that $\bar{z}$ lies in the convex hull of $\mathcal{S}_s^d$. Given our choice of $K$, we have that, with probability $1 - \delta/2 - \beta$

$$\|\hat{z} - \bar{z}\|_2 \lesssim L\left(\frac{[s\ln(d/s)\ln(1/\delta)]^{1/4}}{\sqrt{n\varepsilon}} + \frac{\sqrt{\ln(1/\beta)\ln(1/\delta)}}{n\varepsilon}\right).$$

We conclude by providing the second moment estimate, by a simple tail integration argument. First, by the law of total probability, and letting $\mathcal{E}$ be the event of $A$ being an approximate restricted isometry,

$$\mathbb{E}\|\hat{z} - \bar{z}\|_2^2 \leq \mathbb{E}[\|\hat{z} - \bar{z}\|_2^2 | \mathcal{E}] + 9L^2\delta,$$

where we also used that $\|\hat{z}\|_2 \leq 2L$ and $\|\bar{z}\|_2 \leq L$, almost surely. Now, conditionally on $\mathcal{E}$, we have that letting $\alpha \asymp L\frac{[s\ln(d/s)\ln(1/\delta)]^{1/4}}{\sqrt{n\varepsilon}}$ (below $c > 0$ is an absolute constant),

$$\begin{aligned}
\mathbb{E}[\|\hat{z} - \bar{z}\|_2^2 | \mathcal{E}] &= \int_0^\infty \mathbb{P}\Big[\|\hat{z} - \bar{z}\|_2 \geq u\Big](2u)du \\
&\leq \frac{\alpha^2}{2} + \int_0^\infty \mathbb{P}\Big[\|\hat{z} - \bar{z}\|_2 - \alpha \geq \tau\Big]2(\alpha + \tau)d\tau \\
&\leq \frac{\alpha^2}{2} + \int_0^\infty 2\exp\left\{-\frac{c(n\varepsilon)^2}{L^2\ln(1/\delta)}\tau^2\right\}(\alpha + \tau)d\tau \\
&\lesssim \frac{\alpha^2}{2} + 2\alpha L\frac{\sqrt{\ln(1/\delta)}}{n\varepsilon} + L^2\frac{\ln(1/\delta)}{(n\varepsilon)^2} \\
&\lesssim \alpha^2,
\end{aligned}$$

where in the second inequality we used the previous high probability upper bound (here $c > 0$ is an absolute constant), and in the last step we used that $n\varepsilon > \sqrt{\ln(1/\delta)}$. Finally, by our assumptions on $\delta$, $9L^2\delta \lesssim \alpha^2$, and this concludes the proof. $\square$

# C    Missing Proofs from Section 4

## C.1    Proof of Lemma 4.3

*Proof.* Consider an $n \times d$ data matrix $D$ whose rows correspond to datapoints of a dataset $S$, and whose columns correspond to their $d$ features. We will indistinctively refer to $S$ or $D$ as needed (these are equivalent representations of a dataset). This data matrix will be comprised of $K$ diagonal blocks, $D_1, \dots, D_K$; in particular, outside of these blocks, the matrix has only zeros. These blocks are sampled i.i.d. from the hard distribution $\mu$ given by hypothesis. Denote $\tilde{\mu}$ the law of $D$.

Let now $\bar{z}_k(D_k) \in \mathbb{R}^t$ be the mean (over rows) of dataset $D_k$. Then, the mean (over rows) of dataset $D$ is given by $\bar{z}(D) = \frac{n_0}{n}\big[\bar{z}_1(D_1)\big| \dots \big|\bar{z}_K(D_K)\big]$, where $[z_1| \dots |z_K] \in \mathbb{R}^d$ denotes the concatenation of $z_1, \dots, z_K$ (note that if $K < d/t$, then the concatenation above needs to be padded with $(d - tK)$-zeros, which we omit for simplicity).

Let $\mathcal{A}$ be an $(\varepsilon, \delta)$-DP algorithm, and let $\mathcal{A}_k$ its output on the $k$th block variables, then

$$\|\mathcal{A}(D) - \bar{z}(D)\|_2^2 = \sum_{k=1}^{K} \left\| \mathcal{A}_k(D) - \frac{n_0}{n}\bar{z}_k(D_k) \right\|_2^2 = \frac{n_0^2}{n^2} \sum_{k=1}^{K} \left\| \frac{n}{n_0}\mathcal{A}_k(D) - \bar{z}_k(D_k) \right\|_2^2.$$

Let now $\mathcal{B}_k(D) := \frac{n}{n_0}\mathcal{A}_k(D)$, and note it is $(\varepsilon, \delta)$-DP w.r.t. $D_k$ (as it is DP w.r.t. $D$); further, by the independence of $D_1, \dots, D_K$, we can condition on $(D_h)_{h \neq k}$, to conclude that the squared $\ell_2$-error $\|\mathcal{B}_k(D) - \bar{z}_k(D_k)\|_2^2$ must be at least $\alpha_0^2$, with probability at least $\rho_0$ (both on $D_k$ and the internal randomness of $\mathcal{B}_k$). Letting $Y_k := \mathbf{1}_{\{\|\mathcal{B}_k(D) - \bar{z}_k(D_k)\|_2 \geq \alpha_0\}}$, we have

$$\mathbb{P}\Big[\|\mathcal{A}(D) - \bar{z}(D)\|_2^2 \geq \big(\frac{\alpha_0 n_0}{n}\big)^2 \frac{\rho_0 K}{2}\Big] \geq \mathbb{P}\Big[\sum_{k=1}^{K} Y_k \geq \frac{\rho_0 K}{2}\Big].$$

We will now use a coupling argument to lower bound the probability above. First, we let $U_1, \dots, U_K \overset{i.i.d.}{\sim} \text{Unif}([0,1])$, and $W_k = \mathbf{1}_{\{U_i \geq \rho_0\}}$ which are i.i.d. On the other hand, we define

$$p_k(y_1, \dots, y_{k-1}) := \mathbb{P}[Y_k = 1 | Y_1 = y_1, \dots, Y_{k-1} = y_{k-1}]$$
$$\tilde{Y}_k := \mathbf{1}_{\{U_k \geq p_k(\tilde{Y}_1, \dots, \tilde{Y}_{k-1})\}}.$$

Noting that $Y \overset{d}{=} \tilde{Y}$, and that $\tilde{Y}_k \geq W_k$ almost surely, due to the fact that $p_k \geq \rho_0$ almost surely (which it follows from the $\ell_2$-error argument discussed above), we have

$$\mathbb{P}\Big[\sum_{k=1}^{K} Y_k \geq \frac{\rho_0 K}{2}\Big] = \mathbb{P}\Big[\sum_{k=1}^{K} \tilde{Y}_k \geq \frac{\rho_0 K}{2}\Big] \geq \mathbb{P}\Big[\sum_{k=1}^{K} W_k \geq \frac{\rho_0 K}{2}\Big] \geq 1 - \exp(-\rho_0/8),$$

where we used a one-sided multiplicative Chernoff bound.

Therefore, $\|\mathcal{A}(D) - \bar{z}(D)\|_2^2 \gtrsim \big(\frac{\alpha_0 n_0}{n}\big)^2 \rho_0 K$, with probability $1 - \exp(-\rho_0/8)$. We conclude that $\tilde{\mu}$ induces an $(\alpha, \rho)$-distributional lower bound for $(\varepsilon, \delta)$-DP mean estimation, as claimed. $\qquad\square$

## C.2    Proof of Theorem 4.4

*Proof.* By Lemma A.3, there exists a set $\mathcal{P}$ of $1/\sqrt{2}$-packing vectors on $\mathcal{C}_s^d$ with $\log(|\mathcal{P}|) \gtrsim s \log(d/s)$. Lemma A.5 thus implies the desired lower bound. $\qquad\square$

## C.3    Proof of Theorem 4.1

With all the building blocks in place, we now prove Theorem 4.1.

*Proof of Theorem 4.1.* We divide the analysis into the different regimes of sample size $n$. First, if $n \lesssim \frac{s \log(d/s)}{\varepsilon}$, then Theorem 4.4 provides an $\Omega(1)$ lower bound.

Next we consider the case $\frac{s \log(d/s)}{\varepsilon} \lesssim n \lesssim \frac{d}{\varepsilon}$. For $s \leq t \leq d$ to be determined, let $n_0 = \frac{s \log(t/s)}{\varepsilon}$. We choose $t$ so that $\frac{d}{t} \asymp \frac{n}{n_0}$: this can be attained by choosing $t \asymp \frac{ds}{\varepsilon n} \log\big(\frac{d}{\varepsilon n}\big)$. This implies in the

context of Lemma 4.3 that $K = \frac{d}{t} \asymp \frac{n}{n_0}$. By Theorem 4.4, this implies a lower bound $\alpha_0 \gtrsim 1$, with constant probability $1/2$ for sparse mean estimation in dimension $t$. By Lemma 4.3, we conclude a sparse mean estimation lower bound of $\frac{\alpha_0 n_0}{n}\sqrt{\frac{K}{2}} \gtrsim \frac{1}{\sqrt{K}} \gtrsim \sqrt{\frac{s \log(d/n\varepsilon)}{\varepsilon n}}$ holds with constant probability.

On the other hand, if $n \gtrsim \frac{d}{\varepsilon}$, let $n^* \asymp \frac{d}{\varepsilon}$. By the previous paragraph, for datasets of size $n^*$ the following lower bound holds, $\Omega\left(\sqrt{\frac{s\log(d/\varepsilon n^*)}{\varepsilon n^*}}\right) \gtrsim \sqrt{\frac{s}{d}}$. For any $n > n^*$, by Lemma A.4, we have the lower bound $\Omega\left(\sqrt{\frac{s}{d}}\frac{n^*}{n}\right) \gtrsim \frac{\sqrt{sd}}{\varepsilon n}$ holds with constant probability. $\qquad\square$

## C.4 Proof of Theorem 4.5

*Proof.* We divide the analysis into the different regimes of sample size $n$. First, if $n \lesssim \sqrt{s\ln(1/\delta)}/\varepsilon$, then embedding an $s$-dimensional lower bound construction [42][4] and padding it with zeros for the remaining $d - s$ features, provides an $\Omega(1)$ lower bound with constant probability.

Next, we consider the case $\sqrt{s\ln(1/\delta)}/\varepsilon \lesssim n \lesssim \frac{d\sqrt{\ln(1/\delta)}}{\sqrt{s}\varepsilon}$. Let $n_0 = \sqrt{s\ln(1/\delta)}/\varepsilon$, $t = s$, and $K = \frac{n}{n_0} \lesssim \frac{d}{s}$, where the last inequality holds by our regime assumption. The classic $s$-dimensional mean estimation lower bound by [42] provides an $\alpha_0 \gtrsim 1$ lower bound with constant probability. Hence by Lemma 4.3, the sparse mean estimation problem satisfies a lower bound $\Omega\left(\frac{\alpha_0 n_0}{n}\sqrt{K}\right) \gtrsim \frac{1}{\sqrt{K}} \gtrsim \frac{(s\ln(1/\delta))^{1/4}}{\sqrt{\varepsilon n}}$, with constant probability.

We conclude with the final range, $n \gtrsim \frac{d\sqrt{\ln(1/\delta)}}{\sqrt{s}\varepsilon}$. First, letting $n^* \asymp \frac{d\sqrt{\ln(1/\delta)}}{\sqrt{s}\varepsilon}$, we note that this sample size falls within the range of the previous analysis, which implies a lower bound with constant probability of $\frac{(s\ln(1/\delta))^{1/4}}{\varepsilon\sqrt{n^*}} \gtrsim \frac{\sqrt{s}}{\sqrt{d}}$. Now, if $n > n^*$, by Lemma A.4, we conclude that the following lower bound holds with constant probability, $\Omega\left(\frac{\sqrt{s}}{\sqrt{d}}\frac{n^*}{n}\right) \gtrsim \frac{\sqrt{d\ln(1/\delta)}}{n\varepsilon}$. $\qquad\square$

## D  Analysis of Biased SGD

Given the heavy-tailed nature of our estimators, our guarantees for a single run of SGD with bias-reduced first-order oracles only yields constant probability guarantees. Here we prove pathwise bounds that facilitate such analyses.

### D.1  Excess Empirical Risk: Convex Case

First, we provide a path-wise guarantee for a run of SGD with a biased oracle. Importantly, this guarantee is made of a method which runs for a *random number of steps*.

**Proposition D.1.** *Let $(\mathcal{F}_t)_t$ be the natural filtration, and $T$ be a random time. Let $(x^t)_t$ be the trajectory of projected SGD with deterministic stepsize sequence $(\eta_t)_t$, and (biased) stochastic first-order oracle $\mathcal{G}$ for a given function $F$. If $x^* \in \arg\min\{F(x) : x \in \mathcal{X}\}$, then the following event holds almost surely*

$$\sum_{t=0}^{T}[F(x^t) - F(x^*)] \le \frac{1}{2\eta_t}\|x^0 - x^*\|^2 + \sum_{t=0}^{T}\left[\frac{\eta_t}{2}\|\mathcal{G}(x^t)\|^2 + \langle \nabla F(x^t) - \mathcal{G}(x^t), x^t - x^* \rangle\right].$$

---

[4]While [42] only provides 1-dimensional distributional lower bounds for approximate-DP mean estimation, it is easy to convert these into higher dimensional lower bounds, see, e.g., [26, 43].

*Proof.* By convexity

$$F(x^t) - F(x^*) \leq \langle \nabla F(x^t), x^t - x^* \rangle = \underbrace{\langle \nabla F(x^t) - \mathcal{G}(x^t), x^t - x^* \rangle}_{:=b_t} + \langle \mathcal{G}(x^t), x^t - x^* \rangle$$

$$\leq b_t + \langle \mathcal{G}(x^t), x^t - x^{t+1} \rangle + \langle \mathcal{G}(x^t), x^{t+1} - x^* \rangle$$

$$\leq b_t + \frac{\eta_t}{2} \|\mathcal{G}(x^t)\|^2 + \frac{1}{2\eta_t} \|x^t - x^{t+1}\|^2 + \langle \nabla \mathcal{G}(x^t), x^{t+1} - x^* \rangle$$

$$\overset{(*)}{\leq} b_t + \frac{\eta_t}{2} \|\mathcal{G}(x^t)\|^2 + \frac{1}{2\eta_t} \|x^t - x^{t+1}\|^2 + \frac{1}{\eta_t} \langle x^{t+1} - x^t, x^* - x^{t+1} \rangle$$

$$= b_t + \frac{\eta_t}{2} \|\mathcal{G}(x^t)\|^2 + \frac{1}{2\eta_t} \|x^t - x^*\|^2 - \frac{1}{2\eta_t} \|x^{t+1} - x^*\|^2,$$

where the second inequality follows by the Young inequality, and step $(*)$ we used the optimality conditions of the projected SGD step:

$$\langle \eta_t \mathcal{G}(x^t) + [x^{t+1} - x^t], x - x^{t+1} \rangle \geq 0 \quad (\forall x \in \mathcal{X}).$$

Therefore, summing up these inequalities, we obtain

$$\sum_{t=0}^{T} [F(x^t) - F(x^*)] \leq \frac{1}{2\eta_0} \|x^0 - x^*\|^2 + \sum_{t=0}^{T} \left[ \frac{\eta_t}{2} \|\mathcal{G}(x^t)\|^2 + b_t \right].$$

Plugging in the definition of $b_t$ proves the result. $\qquad\square$

### D.2 Stationary Points: Nonconvex Case

**Proposition D.2.** *Let $F$ satisfy (A.6), and let $\mathcal{G}$ be a biased first-order stochastic oracle for $F$. Let $(x^t)_t$ be the trajectory of SGD with oracle $\mathcal{G}$, constant stepsize $0 < \eta \leq 1/[2H]$, and initialization $x^0$ such that $F(x^0) - \min_{x \in \mathbb{R}^d} F(x) \leq \Gamma$. Let $T$ be a random time. Then the following event holds almost surely*

$$\sum_{t=0}^{T} \|\nabla F(x^t)\|_2^2 \leq \frac{\Gamma}{\eta} + \frac{\eta H}{2} \sum_{t=0}^{T} \|\mathcal{G}(x^t)\|_2^2 - \sum_{t=0}^{T} \langle \nabla F(x^t), \mathcal{G}(x^t) - \nabla F(x^t) \rangle$$

*Proof.* By smoothness of $f$, we have

$$F(x^{t+1}) - F(x^t) \leq -\eta \langle \nabla F(x^t), \mathcal{G}(x^t) \rangle + \frac{\eta^2 H}{2} \|\mathcal{G}(x^t)\|_2^2$$

$$\leq -\eta \|\nabla F(x^t)\|_2^2 - \eta \langle \nabla F(x^t), \mathcal{G}(x^t) - \nabla F(x^t) \rangle + \frac{\eta^2 H}{2} \|\mathcal{G}(x^t)\|_2^2.$$

Therefore,

$$\sum_{t=0}^{T} \|\nabla F(x^t)\|_2^2 \leq \frac{F(x^0) - F(x^{T+1})}{\eta} - \sum_{t=0}^{T} \langle \nabla F(x^t), \mathcal{G}(x^t) - \nabla F(x^t) \rangle + \frac{\eta H}{2} \sum_{t=0}^{T} \|\mathcal{G}(x^t)\|_2^2$$

$$\leq \frac{\Gamma}{\eta} - \sum_{t=0}^{T} \langle \nabla F(x^t), \mathcal{G}(x^t) - \nabla F(x^t) \rangle + \frac{\eta H}{2} \sum_{t=0}^{T} \|\mathcal{G}(x^t)\|_2^2. \qquad\square$$

## E  Missing proofs from Section 5

### E.1  Proof of Lemma 5.1

*Proof.* The proof is based on the fully adaptive composition theorem of DP [15]. For this, we consider $\{\mathcal{A}_t\}_{t \geq 0}$, where $\mathcal{A}_0(S) = (x^0, N_0)$ (here $N_0$ the first truncated geometric parameter), and inductively, $\mathcal{A}_{t+1}(\mathcal{A}_t(S), S)$ for $t \geq 0$ takes as input $\mathcal{A}_t(S) = (x^t, N_t)$, computes $\mathcal{G}(x_t)$ using the subsampled debiased gradient estimator (Algorithm 2), and performs a projected gradient step based on $\mathcal{G}(x^t)$. Let $\mathcal{H}_t$ be the $\sigma$-algebra induced by $(\mathcal{A}_s)_{s=0,\ldots,t}$.

Suppose now that $\mathcal{A}_t$ is $(\varepsilon_t, \delta_t)$-DP, where $(\varepsilon_t, \delta_t)$ are $\mathcal{H}_t$-measurable (we will later obtain these parameters), and let $T := \inf\{t : \varepsilon_{[0:t]} > \varepsilon/2,\ \delta_{[0:t]} > \delta/2\}$, in the language of Theorem A.6 (notice that in the context of that theorem, we are choosing $\delta' = \delta'' = \delta/4$). We first claim that $(x^t)_{t=0,\ldots,T-1}$ is $(\varepsilon/2, \delta/2)$-DP, which follows directly from Theorem A.6. Next, we will later show that $\varepsilon_t \le \varepsilon/4$ and $\delta_t \le \delta/4$, almost surely (this applies in particular to $x_T$), and therefore by the composition property of DP, $(x_t)_{t \le T}$ is $(\varepsilon, \delta)$-DP.

Next, we provide the bounds on $(\varepsilon_t, \delta_t)$ required to conclude the proof. For this, we first note that—conditionally on $x^t$, $N_t$ and $B_t$—the computation of $G^+_{N_t+1}(x^t, B_t)$, $G^-_{N_t}(x^t, O_t)$, $G^-_{N_t}(x^t, E_t)$, is $(3\varepsilon/32, 3\delta/16)$-DP. Furthermore, by privacy amplification by subsampling, this triplet of random variables is $(\varepsilon', \delta')$, with

$$\varepsilon' = \ln\left(1 + \frac{2^{N_t+1}}{n}(e^{3\varepsilon/32} - 1)\right) \le \frac{2^{N_t+1}}{n}\frac{3\varepsilon}{16}, \qquad \delta' = \frac{2^{N_t+1}}{n}\frac{3\delta}{16},$$

where we used above that $\varepsilon \le 1$. Similarly, we have that $G_0(x, I)$ is $\left(\frac{\varepsilon}{16n}, \frac{\delta}{16n}\right)$-DP. Therefore, by the basic composition theorem of DP, we have the following privacy parameters for the $t$th iteration of the algorithm

$$\varepsilon_t = (3 \cdot 2^{N_t+1} + 1)\frac{\varepsilon}{16n}, \quad \delta_t = (3 \cdot 2^{N_t+1} + 1)\frac{\delta}{16n}.$$

This proves in particular that $(\varepsilon_t, \delta_t)$ are $\mathcal{H}_t$-measurable, and that $\varepsilon_t \le \varepsilon/4$, and $\delta_t \le \delta/4$ almost surely, which concludes the proof □

### E.2 Proof of Lemma 5.3

*Proof.* Let $A = \sum_{s=0}^{t-1}\left(\frac{3 \cdot 2^{N_s+1}+1}{16n}\right)^2$, and note that for $t \le T+1$, $A \le 1$ almost surely. Then, we have that

$$\varepsilon_{[0:t-1]} = \sqrt{2\ln(4/\delta)\varepsilon^2 A} + \frac{\varepsilon^2}{2}A \le 2\varepsilon\sqrt{2\ln(4/\delta)A}.$$

Now, by eqn. (2) and the union bound,

$$\mathbb{P}[T \le t] \le \mathbb{P}\left[2\varepsilon\sqrt{2\ln(4/\delta)A} > \varepsilon/2\right] + \mathbb{P}\left[\sum_{s=0}^{t-1}(3 \cdot 2^{N_t+1} + 1) > 4n\right]$$

$$\le \mathbb{P}\left[\sum_{s=0}^{t-1}\left(3 \cdot 2^{N_t+1} + 1\right)^2 > \frac{32n^2}{\ln\left(\frac{4}{\delta}\right)}\right] + \mathbb{P}\left[\sum_{s=0}^{t-1}(3 \cdot 2^{N_t+1} + 1) > 4n\right]$$

$$\le \frac{t\ln\left(\frac{4}{\delta}\right)}{16n^2}\left(9\mathbb{E}[2^{2(N_t+1)}] + 1\right) + \frac{t}{4n}[6(M+1) + 1]$$

$$\le \frac{t\ln\left(\frac{4}{\delta}\right)}{16n^2}[18n + 1] + \frac{t}{4n}[6\log(n) + 1]$$

$$\le 1/4,$$

where the third step follows from Markov's inequality and the fact that $(N_s)_s$ are i.i.d., and the last step follows from our choice of $t = Cn/\log(4/\delta)$ with $C > 0$ sufficiently small (here we use the fact that $\delta < 1/n^2$).

For the second part, we use that by the definition of $T$ (eqn. (2))

$$\frac{\varepsilon}{2} < \sqrt{2\varepsilon^2\ln\left(\frac{4}{\delta}\right)\sum_{s=0}^{T}\frac{(3 \cdot 2^{N_s+1}+1)^2}{(16n)^2} + \frac{\varepsilon^2}{2}\sum_{s=0}^{T}\frac{(3 \cdot 2^{N_s+1}+1)^2}{(16n)^2}} \quad \vee \quad \frac{1}{4} < \sum_{s=0}^{T}\frac{3 \cdot 2^{N_s+1}+1}{16n}$$

$$\implies \quad n^2 < \max\left\{8\ln\left(\frac{4}{\delta}\right)\sum_{s=0}^{T}\frac{(3 \cdot 2^{N_s+1}+1)^2}{(16)^2}, n\sum_{s=0}^{T}\frac{3 \cdot 2^{N_s+1}+1}{4}\right\}$$

Taking expectations and bounding the maximum by the sum allows us to use Wald's identity as follows,

$$n^2 < \mathbb{E}[T+1]\left(8\ln\left(\frac{4}{\delta}\right)\frac{2(9n+1)}{16^2} + n\frac{3\log(n)+1}{4}\right)$$

$$\le \mathbb{E}[T+1]\ln\left(\frac{4}{\delta}\right)(n+1),$$

which proves the claimed bound.

The upper nound on $\mathbb{E}[T]$ is obtained similarly. Again, by eqn. (2),

$$\frac{32n^2}{\ln(4/\delta)} \geq \mathbb{E}\Big[\sum_{s=0}^{T-1} \big(3 \cdot 2^{N_s+1} + 1\big)^2\Big] \geq \mathbb{E}[T]\frac{9n}{2}.$$

Re-arranging terms provides the claimed lower bound. $\qquad\square$

### E.3 Excess Empirical Risk in the Convex Setting

As a first application, we study the accuracy guarantees of Algorithm 3 in the convex setting. We remark that these rates will be slightly weaker than those provided in Section 6, but this example is useful to illustrate the technique. Towards this goal, we analyze the cumulative regret of the algorithm, namely $\mathcal{R}_T := \sum_{t=0}^{T}[F_S(x^t) - F_S(x^*(S))]$. Although this is a standard and well-studied object in optimization, we need to obtain bounds for this object when the stopping time $T$ is random. The key observation here is that since $T$ is a stopping time, the event $\{T \geq t\}$ is $\mathcal{F}_{t-1}$-measurable (here and throughout, $\mathcal{F}_t = \sigma((x_s)_{s \leq t})$ is the natural filtration). This permits using our bias and second moment bounds similarly to the case where $T$ is deterministic.[5] Moreover, for the sake of analysis, we will consider Algorithm 3 as running indefinitely, for all $t \geq 0$. This would of course eventually violate privacy. However, since our algorithm stops at time $T$, then privacy is guaranteed as done earlier in this section.

**Proposition E.1.** *Let $\mathcal{R}_t := \sum_{t=0}^{t}[F_S(x^t) - F_S(x^*(S))]$, let $T$ be the stopping time defined in eqn. (2). Then*

$$\mathbb{E}[\mathcal{R}_T] \leq \frac{1}{2\eta}\|x^0 - x^*(S)\|^2 + \mathbb{E}[T+1]\big(\frac{\eta\nu^2}{2} + Db\big),$$

*where $b$ and $\nu^2$ are defined as in Lemma 5.2.*

*Proof.* By Proposition D.1 (see Appendix D),

$$\mathbb{E}[\mathcal{R}_T]$$
$$\leq \mathbb{E}\Big(\frac{1}{2\eta}\|x^0 - x^*(S)\|^2 + \sum_{t=0}^{T}\big[\frac{\eta}{2}\|\mathcal{G}(x^t)\|^2 + \langle\nabla F(x^t) - \mathcal{G}(x^t), x^t - x^*(S)\rangle\big]\Big)$$
$$= \mathbb{E}\Big(\frac{1}{2\eta}\|x^0 - x^*(S)\|^2$$
$$+ \sum_{t=0}^{\infty}\Big\{\frac{\eta}{2}\mathbb{E}[\mathbf{1}_{\{T \geq t\}}\|\mathcal{G}(x^t)\|^2|\mathcal{F}_{t-1}] + \mathbb{E}[\mathbf{1}_{\{T \geq t\}}\langle\nabla F(x^t) - \mathcal{G}(x^t), x^t - x^*(S)\rangle|\mathcal{F}_{t-1}]\Big\}\Big)$$
$$= \mathbb{E}\Big(\frac{1}{2\eta}\|x^0 - x^*(S)\|^2$$
$$+ \sum_{t=0}^{\infty}\Big\{\frac{\eta\mathbf{1}_{\{T \geq t\}}}{2}\mathbb{E}[\|\mathcal{G}(x^t)\|^2|\mathcal{F}_{t-1}] + \mathbf{1}_{\{T \geq t\}}\mathbb{E}[\langle\nabla F(x^t) - \mathcal{G}(x^t), x^t - x^*(S)\rangle|\mathcal{F}_{t-1}]\Big\}\Big)$$

where in the first equality we used the tower property of the conditional expectation, and in the second equality we used that $\{T \geq t\} = \{T \leq t-1\}^c$ is $\mathcal{F}_{t-1}$-measurable.

Now, by Lemma 5.2, $\mathbb{E}[\langle\nabla F(x^t) - \mathcal{G}(x^t), x^t - x^*(S)\rangle|\mathcal{F}_{t-1}] \leq Db$ and $\mathbb{E}[\|\mathcal{G}(x^t)\|^2|\mathcal{F}_{t-1}] \leq \nu^2$ (note that $\mathcal{F}_{t-1}$ does not include the randomness of $N_t$, and therefore the bias and moment estimates as in the mentioned lemma hold), thus

$$\mathbb{E}[\mathcal{R}_T] \leq \frac{1}{2\eta}\|x^0 - x^*(S)\|^2 + \mathbb{E}[T+1]\big(\frac{\eta\nu^2}{2} + Db\big). \qquad\square$$

We conclude with the constant probability guarantee for the biased and randomly stopped SGD, Algorithm 3.

---

[5]This idea is related to the Wald identities [16]; however, we provide a direct analysis for the sake of clarity.

**Theorem E.2.** *Consider a* (SO) *problem under convexity (Item (A.3)), initial distance (Item (A.1)), Lipschitzness (Item (A.5)) and gradient sparsity (Item (A.7)) assumptions. Let $\tau = \frac{C'n}{\ln(2/\delta)}$, where $C' > 0$ is an absolute constant. Let $\eta = \frac{D}{\nu\sqrt{\tau}}$, $U = CD[\nu\sqrt{\tau} + b\tau]$, where $C > 0$ is an absolute constant. Then Algorithm 3 satisfies*

$$\mathbb{P}\Big[F_S(\bar{x}) - F_S(x^*(S)) \le \frac{U}{\tau}\Big] \ge 1/2.$$

*Proof.* We start by noting that

$$\mathbb{P}\Big[F_S(\bar{x}) - F_S(x^*(S)) > \frac{U}{\tau}\Big] \le \mathbb{P}\big[\{T \le \tau\} \cup \{\mathcal{R}_T > U\}\big] \le \mathbb{P}\big[T \le \tau\big] + \mathbb{P}[\mathcal{R}_T > U].$$

For the first event, by Lemma 5.3, we have that $\mathbb{P}[T \le \tau] \le 1/4$ (which determines $C'$). On the other hand, using Proposition E.1 and Lemma 5.3, we have that for our choice of $\eta$, we have that

$$\mathbb{E}[\mathcal{R}_T] \le \frac{D\nu\sqrt{\tau}}{2} + \mathbb{E}[T+1]D\Big(\frac{\nu}{2\sqrt{\tau}} + b\Big) \lesssim D[\nu\sqrt{\tau} + \tau b].$$

In particular, for our choice of $U$ (with $C > 0$ sufficiently large),

$$\mathbb{P}[\mathcal{R}_T > U] \le \frac{\mathbb{E}[\mathcal{R}_T]}{U} \le \frac{1}{4}. \qquad \square$$

The above result implies a nearly optimal empirical excess risk rate for DP-SCO,

$$O\Big(LD\,\frac{\sqrt{\ln n}[s\ln(d/s)\ln^3(1/\delta)]^{1/4}}{\sqrt{\varepsilon n}}\Big),$$

but only with constant probability. We defer to the next section how to boost this guarantee to hold with arbitrarily high probability.

### E.3.1 Near Stationary Points for the Empirical Risk

For nonconvex objectives it is known that obtaining vanishing excess risk is computationally difficult. Hence, we study the more modest goal of approximating stationary points, i.e., points with small norm of the gradient. By combining known analyses of biased SGD with our bias-reduced oracle, we can establish bounds on the success probability of the algorithm.

**Theorem E.3.** *Consider a (nonconvex)* (SO) *problem, under the following assumptions: Lipschitzness (Item (A.5)), smoothness (Item (A.6)), gradient sparsity (Item (A.7)), and the following initial suboptimality assumption: namely, that given our initialization $x^0 \in \mathbb{R}^d$, we know $\Gamma > 0$ such that*

$$F_S(x^0) - F_S(x^*(S)) \le \Gamma. \tag{5}$$

*Let $\tau = \frac{C'n}{\ln(2/\delta)}$ with $C' > 0$ an absolute constant. Let $\eta = \sqrt{\frac{\Gamma}{Ht\nu^2}}$ and $U = C\big(\sqrt{\Gamma H\tau}\nu + L\tau b\big)$ with $C > 0$ an absolute constant. Then Algorithm 3 satisfies $\mathbb{P}\Big[\|\nabla F_S(x^{\hat{t}})\|_2^2 \le \frac{U}{\tau}\Big] \ge 1/2$, and*

$$\frac{U}{\tau} \lesssim \big(\sqrt{\Gamma H}L\sqrt{\ln(n)\ln(1/\delta)} + L^2\big)\frac{[s\ln(d/s)\ln(1/\delta)]^{1/4}}{\sqrt{\varepsilon n}}.$$

*Proof.* First, given any $U > 0$, we have that

$$\mathbb{P}\Big[\|\nabla F_S(x_{\hat{t}})\|_2 > \sqrt{\frac{U}{\tau}}\Big] \le \mathbb{P}[T < \tau] + \mathbb{P}[T\|\nabla F_S(x_{\hat{t}})\|_2^2 > U] \le \frac{1}{4} + \frac{\mathbb{E}[T\|\nabla F_S(x^{\hat{t}})\|_2^2]}{U},$$

where the last step follows by Lemma 5.3 and Chebyshev's inequality, respectively. Next, by definition of $\hat{t}$ and Proposition D.2 (see Appendix D.2),

$$\mathbb{E}[(T+1)\|\nabla F(x^{\hat{t}})\|_2^2] = \mathbb{E}\Big[\sum_{t=0}^{T}\|\nabla F(x^t)\|_2^2\Big]$$

$$\leq \frac{\Gamma}{\eta} + \frac{\eta H}{2}\mathbb{E}\Big[\sum_{t=0}^{T}\|\mathcal{G}(x^t)\|_2^2\Big] - \mathbb{E}\Big[\sum_{t=0}^{T}\langle\nabla F(x^t),\mathcal{G}(x^t)-\nabla F(x^t)\rangle\Big]$$

$$\leq \frac{\Gamma}{\eta} + \frac{\eta H}{2}\sum_{t=0}^{\infty}\mathbb{E}[\mathbf{1}_{\{T\geq t\}}\|\mathcal{G}(x^t)\|_2^2] - \sum_{t=0}^{\infty}\mathbb{E}[\mathbf{1}_{\{T\geq t\}}\langle\nabla F(x^t),\mathcal{G}(x^t)-\nabla F(x^t)\rangle]$$

$$\leq \frac{\Gamma}{\eta} + \frac{\eta H}{2}\sum_{t=0}^{\infty}\mathbb{P}[T\geq t]\mathbb{E}\big(\mathbb{E}[\|\mathcal{G}(x^t)\|_2^2|\mathcal{F}_{t-1}]\big)$$

$$-\sum_{t=0}^{\infty}\mathbb{P}[T\geq t]\mathbb{E}\big(\mathbb{E}[\langle\nabla F(x^t),\mathcal{G}(x^t)-\nabla F(x^t)|\mathcal{F}_{t-1}\rangle]\big)$$

$$\leq \frac{\Gamma}{\eta} + \frac{\eta H}{2}\mathbb{E}[T+1]\nu^2 + \mathbb{E}[T+1]Lb$$

$$\lesssim \sqrt{\Gamma H\tau}\nu + \tau Lb,$$

where the third inequality holds since $\{T \geq t\}$ is $\mathcal{F}_{t-1}$-measurable (see the proof of Theorem E.1 for details), and the fourth inequality follows from Theorem 5.2, used the upper bound on $\mathbb{E}[T]$ from Lemma 5.3, and our choice for $\eta$. Selecting $U = C\big(\sqrt{\Gamma H\tau}\nu + L\tau b\big)$ with $C > 0$ sufficiently large, we get $\mathbb{E}[T\|\nabla F(x^{\hat{t}})\|_2^2]/U \leq 1/4$, concluding the proof. $\qquad\square$

## F  Boosting the Confidence for the Bias-Reduced Stochastic Gradient Method

We conclude by providing a boosting method to amplify the success probability of our bias-reduced method. This private boosting method is a particular instance of a private selection method [17], and it is based on running a random number of independent runs of Algorithm 3 with noisy evaluations of their performance. Among the independent runs, we select the best performing one based on the noisy evaluations. This particular implementation sharpens some polylogarithmic factors that would appear for other private selection methods, such as Report Noisy Min [18, 1].

---

**Algorithm 6** `Boosting_Bias-Reduced_SGD`$(S, \varepsilon, \delta, K)$

---

**Require:** Dataset $S \sim \mathcal{D}^n$, $\varepsilon, \delta > 0$ privacy parameters, random stopping parameter $\gamma \in (0,1)$
$\quad K = \frac{1}{\gamma}\ln\big(\frac{2}{\delta}\big)$
$\quad$**for** $k = 1, \ldots, K$ **do**
$\quad\quad$ Run Algorithm 3 with privacy budget $(\varepsilon/12, (\delta/[4K])^2)$, $\hat{x}_k$ its output and
$\quad\quad$**if** $f(\cdot, z)$ convex **then**
$\quad\quad\quad$ Set $s_k = [F_S(\hat{x}_k) + \xi_k]$, where $\xi_k \sim \mathsf{Lap}(\lambda)$, and $\lambda = \frac{12B}{n\varepsilon}$.
$\quad\quad$**else**
$\quad\quad\quad$ Set $s_k = [\|\nabla F_S(\hat{x}_k)\|_2 + \xi_k]$, where $\xi_k \sim \mathsf{Lap}(\lambda)$, and $\lambda = \frac{24L}{n\varepsilon}$.
$\quad\quad$**end if**
$\quad\quad$ Flip a $\gamma$-biased coin: with probability $\gamma$, **return** $\hat{x} = \hat{x}_{\hat{k}}$, where $\hat{k} = \arg\min_{l\leq k} s_l$
$\quad$**end for**
$\quad$**Return** $\hat{x} = \hat{x}_{\hat{K}}$, where $\hat{K} = \arg\min_{k\leq K} s_k$

---

**Theorem F.1.** *Let $\varepsilon, \delta > 0$ such that $\delta \leq \varepsilon/10$. Then Algorithm 6 is $(\varepsilon, \delta)$-DP. Let $0 < \beta < 1$ and $\gamma = \min\{1/2, 3\beta/4\}$. In the convex case, Algorithm 6 attains excess risk $\mathbb{P}\Big[F_S(\hat{x}) - F_S(x^*(S)) \leq \alpha\Big] \geq 1 - \beta$, where*

$$\alpha \lesssim LD\frac{\sqrt{\ln n}[s\ln(d/s)\ln^3\big(\ln(1/\delta)/[\beta\delta]\big)]^{1/4}}{\sqrt{\varepsilon n}} + \frac{B}{n\varepsilon}\ln\Big(\frac{1}{\beta}\ln\big(\frac{2}{\delta}\big)\Big).$$

*On the other hand, in the nonconvex case,* $\mathbb{P}\Big[\|\nabla F_S(\hat{x})\|_2^2 \leq \alpha\Big] \geq 1 - \beta,$ *where*

$$\alpha \lesssim \Big(\sqrt{\Gamma H} L \sqrt{\ln(n)\ln\Big(\frac{\ln(1/\delta)}{\beta\delta}\Big)} + L^2\Big)\frac{[s\ln(d/s)\ln(\ln(1/\delta)/[\beta\delta])]^{1/4}}{\sqrt{\varepsilon n}} + \frac{L}{n\varepsilon}\ln\Big(\frac{1}{\beta}\ln\Big(\frac{2}{\delta}\Big)\Big).$$

*Proof.* The privacy analysis follows easily from [17]. First, by basic composition, we have that for each $k$ the pair $(\hat{x}_k, s_k)$ is $(\varepsilon_1, \delta_1)$-DP, with $\varepsilon_1 = \varepsilon/6$, and $\delta_1 = (\delta/[4K])^2$. By [17, Thm 3.4], the private selection with random stopping used in Algorithm 6 is such that $\hat{x}$ is $(3\varepsilon_1 + 3\sqrt{2\delta_1}, \sqrt{2\delta_1}K + \delta/2)$-DP; notice that

$$3\varepsilon_1 + 3\sqrt{2\delta_1} \leq \frac{\varepsilon}{2} + 3\sqrt{2}\frac{\delta}{K} \leq \varepsilon,$$

and

$$\sqrt{2\delta_1}K + \delta/2 \leq \delta,$$

due to our choices of $\varepsilon_1, \delta_1$. This proves that the algorithm is $(\varepsilon, \delta)$-DP.

The accuracy of the algorithm closely follows [17, Theorem 3.3]. First, let $\kappa$ be the number of runs the algorithm makes before stopping, and let $\alpha > 0$ to be determined. Conditioning on $\kappa$

$$\mathbb{P}\big[F_S(\hat{x}) - F_S(x^*(S)) > \alpha\big] = \sum_{k=1}^{K} \mathbb{P}\big[F_S(\hat{x}) - F_S(x^*(S)) > \alpha \big| \kappa = k\big]\mathbb{P}[\kappa = k]$$

$$= \sum_{k=1}^{K} \mathbb{P}\big[F_S(\hat{x}) - F_S(x^*(S)) > \alpha \big| \kappa = k\big](1-\gamma)^{k-1}\gamma.$$

We will now bound the conditional probability above. By the subexponential tails of the Laplace distribution, we have that letting $\mathcal{E} := \{(\forall j \in [\kappa]) : |\xi_j| \leq \alpha'\}$ (here, $\alpha' > 0$ is arbitrary),

$$\mathbb{P}[\mathcal{E}^c | \kappa = k] = \mathbb{P}\Big[(\exists j \in [\kappa]) |\xi_k| > \alpha' \Big| \kappa = k\Big] \leq 2k\exp\Big\{-\frac{n\varepsilon\alpha'}{12B}\Big\}.$$

Hence

$$\mathbb{P}\Big[F_S(\hat{x}) - F_S(x^*(S)) > \alpha \Big| \kappa = k\Big] \leq \mathbb{P}\Big[\{F_S(\hat{x}) - F_S(x^*(S)) > \alpha\} \cap \mathcal{E} \Big| \kappa = k\Big] + \mathbb{P}[\mathcal{E}^c | \kappa = k].$$

Next we have

$$\mathbb{P}\Big[\{F_S(\hat{x}) - F_S(x^*(S)) > \alpha\} \cap \mathcal{E} \Big| \kappa = k\Big] \leq \mathbb{P}\Big[\{F_S(\hat{x}_{\hat{k}}) + \xi_{\hat{k}} - F_S(x^*(S)) > \alpha - \alpha'\} \cap \mathcal{E} \Big| \kappa = k\Big]$$

$$= \mathbb{P}\Big[\{\min_{k \in [\kappa]}\big[F_S(\hat{x}_k) + \xi_k\big] - F_S(x^*(S)) > \alpha - \alpha'\} \cap \mathcal{E} \Big| \kappa = k\Big]$$

$$\leq \mathbb{P}\Big[\min_{k \in [\kappa]}\big[F_S(\hat{x}_k) - F_S(x^*(S))\big] > \alpha - 2\alpha' \Big| \kappa = k\Big]$$

$$\leq \Big(\mathbb{P}\Big[F_S(\hat{x}_1) - F_S(x^*(S)) > \alpha - 2\alpha'\Big]\Big)^k,$$

where in the last step we used that the runs are i.i.d.

We now choose $\alpha, \alpha'$ such that $\alpha - 2\alpha' = U/\tau$ (where $U, \tau$ are those from Theorem E.2). Hence,

$$\mathbb{P}\Big[F_S(\hat{x}) - F_S(x^*(S)) > \alpha \Big| \kappa = k\Big] \leq 2^{-k} + 2k\exp\Big\{-\frac{n\varepsilon\alpha'}{12B}\Big\}.$$

We can now bound the failure probability as follows:

$$\mathbb{P}\big[F_S(\hat{x}) - F_S(x^*(S)) > \alpha\big] \leq \sum_{k=1}^{K}\Big(2^{-k} + 2K\exp\Big\{-\frac{n\varepsilon\alpha'}{12B}\Big\}\Big)(1-\gamma)^{k-1}\gamma$$

$$= \frac{1}{2}\frac{\gamma}{1-\gamma^2} + \frac{2}{\gamma}\ln\Big(\frac{2}{\delta}\Big)\exp\Big\{-\frac{n\varepsilon\alpha'}{12B}\Big\}$$

$$\leq \frac{\beta}{2} + \frac{2}{\gamma}\ln\Big(\frac{2}{\delta}\Big)\exp\Big\{-\frac{n\varepsilon\alpha'}{12B}\Big\},$$

where in the last step we used that $\gamma = \min\{1/2, 3\beta/4\}$. It is clear then that $\alpha' = \frac{12B}{n\varepsilon} \ln\left(\frac{16}{3\beta^2} \ln\left(\frac{2}{\delta}\right)\right)$ makes the probability above at most $\beta$. These choices lead to a final bound

$$\alpha = \frac{U}{\tau} + 2\alpha' \lesssim LD \frac{\sqrt{\ln n}[s\ln(d/s)\ln^3\left(\ln(1/\delta)/[\beta\delta]\right)]^{1/4}}{\sqrt{\varepsilon n}} + \frac{B}{n\varepsilon}\ln\left(\frac{1}{\beta}\ln\left(\frac{2}{\delta}\right)\right).$$

For the nonconvex case, we need to replace $B$ by $2L$ in the Laplace concentration bound. Further, we consider the event $\{\|\nabla F(\hat{x}_k)\|_2 > \alpha\}$ (as opposed to the optimality gap event). This implies that we need to set $\alpha > 0$ such that $\alpha - 2\alpha' \geq \sqrt{U/\tau}$ from Theorem E.3. This leads to

$$\mathbb{P}\Big[\|F_S(\hat{x})\|_2 > \alpha\Big] \leq \sum_{k=1}^{K}\left(2^{-k} + 2K\exp\left\{-\frac{n\varepsilon\gamma}{24L}\right\}\right)(1-\gamma)^{k-1}\gamma.$$

The rest of the derivations are analogous. $\qquad\square$

# G    Missing Proofs and Results from Section 6

## G.1    Proof of Theorem 6.1

*Proof.* We proceed by cases:

- **Case $\delta = 0$.** First, we prove that privacy of the algorithm. To do this, we first establish a bound on the $\ell_1$-sensitivity of the (quadratically) regularized ERM. Note that the first-order optimality conditions in this case correspond to

$$x_\lambda^*(S) = -\frac{1}{\lambda}\nabla F_S(x_\lambda^*(S)).$$

Therefore, if $S \simeq S'$, where $S = (z_1, \ldots, z_n)$ and $S = (z_1', \ldots, z_n')$ only differ in one entry,

$$\begin{aligned}
\|x_\lambda^*(S) - x_\lambda^*(S')\|_1 &\leq \frac{1}{\lambda}\|\nabla F_S(x_\lambda^*(S)) - \nabla F_{S'}(x_\lambda^*(S'))\|_1 \\
&\leq \frac{1}{\lambda n}\sum_{i=1}^{n}\|\nabla f(x_\lambda^*(S), z_i) - \nabla f(x_\lambda^*(S'), z_i')\|_1 \\
&\leq \frac{1}{\lambda n}\Big[(n-1)\sqrt{2s}H\|x_\lambda^*(S) - x_\lambda^*(S')\|_2 + 2\sqrt{2s}L\Big] \\
&\leq \frac{1}{\lambda n}\Big(4\sqrt{2s}HL\frac{n-1}{\lambda n} + 2\sqrt{2s}L\Big) \\
&\leq \frac{2\sqrt{2s}L}{\lambda n}\Big(\frac{2H}{\lambda} + 1\Big).
\end{aligned}$$

Above, in the third inequality we used the gradient sparsity (A.7), and the smoothness (A.6), assumptions. In the fourth inequality we used that the regularized ERM has $\ell_2$-sensitivity $\frac{4L}{\lambda n}$ [44, 45, 46]. We conclude the privacy then by Theorem A.1(a).

We also remark that by Theorem A.2(a)-(i), $\|\xi\|_\infty \lesssim \frac{L\sqrt{s}\ln(d/\beta)}{\lambda n\varepsilon}\left(\frac{H}{\lambda} + 1\right)$, with probability $1 - \beta$.

- **Case $\delta > 0$.** The privacy guarantee follows from the fact that the $\ell_2$-sensitivity of $x_\lambda^*(S)$ is $\frac{4L}{\lambda n}$ [44, 45, 46], together with Theorem A.1(b).

Moreover, by Theorem A.2(b)-(i), $\|\xi\|_\infty \lesssim \frac{L\sqrt{\ln(d/\beta)}}{\lambda n\varepsilon}$, with probability $1 - \beta$.

We continue with the accuracy analysis, making a unified presentation for both pure and approximate-DP. First, by the optimality conditions of the regularized ERM,

$$F_S(x_\lambda^*(S)) - F_S(x^*(S)) \leq \frac{\lambda}{2}\|x^*(S)\|^2 \leq \frac{\lambda}{2}D^2. \tag{6}$$

We need the following key fact, which follows by the definitions of $\hat{x}$ and $\tilde{x}$,

$$\|\hat{x} - x_\lambda^*(S)\|_\infty \leq \|\hat{x} - \tilde{x}\|_\infty + \|\tilde{x} - x_\lambda^*(S)\|_\infty \leq 2\|\xi\|_\infty. \tag{7}$$

Using these two bounds, we proceed as follows

$$F_S(\hat{x}) - F_S(x^*(S)) \leq F_S(\hat{x}) - F_S(x_\lambda^*(S)) + \frac{\lambda}{2}D^2 \leq \langle \nabla F_S(\hat{x}), \hat{x} - x_\lambda^*(S)\rangle + \frac{\lambda}{2}D^2$$

$$\leq \|\nabla F_S(\hat{x})\|_1 \|\hat{x} - x_\lambda^*(S)\|_\infty + \frac{\lambda}{2}D^2$$

$$\leq \sqrt{2s}L\|\xi\|_\infty + \frac{\lambda}{2}D^2,$$

where the second inequality follows by convexity of $F_S$, and the fourth one by the gradient sparsity assumption and (7).

The conclusion follows by plugging in the respective bounds of $\lambda$ and $\|\xi\|_\infty$, for both pure- and approximate-DP cases.

$\square$

## G.2 Proof of Theorem 6.3

**Remark G.1.** *Note first that in the proof below we are not addressing the privacy of Algorithm 4, as this has already been proven in Theorem 6.1.*

*On the other hand, note that the same proof below—using the in-expectation generalization guarantees of uniformly stable algorithms [44]— provides a sharper upper bound for the expected excess risk for the pure and approximate-DP cases, which would hold w.p. $1 - \beta$ over the algorithm internal randomness*

$$\mathbb{E}_S[F_\mathcal{D}(\hat{x}) - F_\mathcal{D}(x^*(\mathcal{D}))] \lesssim L^{2/3}H^{1/3}D^{4/3}\left(\frac{s\log(d/\beta)}{\varepsilon n}\right)^{1/3},$$

$$\mathbb{E}_S[F_\mathcal{D}(\hat{x}) - F_\mathcal{D}(x^*(\mathcal{D}))] \lesssim LD\frac{[s\ln(1/\delta)\log(d/\beta)]^{1/4}}{\sqrt{\varepsilon n}}.$$

*Proof.* Using the $\ell_2$-sensitivity of $x_\lambda^*(S)$, $\Delta_2 = \frac{4L}{\lambda n}$, we have the following generalization bound [47]: with probability $1 - \beta/2$

$$F_\mathcal{D}(x_\lambda^*(S)) - F_S(x_\lambda^*(S)) \lesssim \frac{L^2}{\lambda n}\ln(n)\ln\left(\frac{1}{\beta}\right) + B\sqrt{\frac{\ln\left(\frac{1}{\beta}\right)}{n}} =: \gamma.$$

The bound of (6) can be obviously modified by comparison with the population risk minimizer, $x^*(\mathcal{D})$: in particular, the event above[6] implies that

$$F_\mathcal{D}(x_\lambda^*(S)) - F_\mathcal{D}(x^*(\mathcal{D})) \lesssim F_S(x_\lambda^*(S)) - F_S(x^*(\mathcal{D})) + \gamma \leq \frac{\lambda}{2}\|x^*(\mathcal{D})\|_2^2 + \gamma \lesssim \lambda D^2 + \gamma.$$

On the other hand, the bound (7) works exactly as in the proof of Theorem 6.1. Hence, we have that with probability $1 - \beta/2$,

$$F_\mathcal{D}(\hat{x}) - F_\mathcal{D}(x^*(\mathcal{D})) \lesssim F_\mathcal{D}(\hat{x}) - F_\mathcal{D}(x_\lambda^*(S)) + \lambda D^2 + \gamma$$

$$\lesssim \langle \nabla F_\mathcal{D}(\hat{x}), \hat{x} - x_\lambda^*(S)\rangle + \lambda D^2 + \gamma$$

$$\lesssim 2L\sqrt{s}\|\xi\|_\infty + \frac{L^2}{\lambda n}\ln(n)\ln\left(\frac{1}{\beta}\right) + \lambda D^2 + \frac{B}{\sqrt{n}}\sqrt{\ln\left(\frac{1}{\beta}\right)},$$

where in the last step we used that $\|\nabla F_\mathcal{D}(\hat{x})\|_1 = \|\mathbb{E}_z[\nabla f(\hat{x}, z)]\|_1 \leq \mathbb{E}_z[\|\nabla f(\hat{x}, z)\|_1] \leq L\sqrt{s}$ (the last step which follows by the gradient sparsity), inequality (7), and the definition of $\gamma$.

We proceed now by separately studying the different cases for $\delta$:

---

[6]We also need concentration to upper bound $F_S(x^*(\mathcal{D})) - F_\mathcal{D}(x^*(\mathcal{D}))$. However, this is easy to do by e.g., Hoeffding's inequality, leading to a bound $\lesssim \gamma$.

- **Case $\delta = 0$.** The bound above becomes

$$F_{\mathcal{D}}(\hat{x}) - F(x^*(\mathcal{D})) \lesssim \frac{L^2}{\lambda n}\Big(\frac{s\ln(d/\beta)}{\varepsilon}\big(\frac{H}{\lambda} + 1\big) + \ln n \ln(1/\beta)\Big) + \lambda D^2 + B\sqrt{\frac{\ln(1/\beta)}{n}}.$$

Our choice of $\lambda$ provides the claimed bound.

- **Case $\delta > 0$.** Here, the upper bound takes the form

$$F_{\mathcal{D}}(\hat{x}) - F(x^*(\mathcal{D})) \lesssim \frac{L^2}{\lambda n}\Big(\frac{\sqrt{s\ln(d/\beta)\ln(1/\delta)}}{\varepsilon} + \ln(n)\ln(1/\beta)\Big) + \lambda D^2 + B\sqrt{\frac{\ln(1/\beta)}{n}}.$$

The proposed value of $\lambda$ leads to the bound below that holds with probability $1 - \beta$,

$$F_{\mathcal{D}}(\hat{x}) - F_{\mathcal{D}}(x^*(\mathcal{D})) \lesssim B\sqrt{\frac{\ln(1/\beta)}{n}} + LD\sqrt{\frac{\ln n \ln(1/\beta)}{n}} + \frac{\sqrt{s\ln(1/\delta)\log(d/\beta)}}{\varepsilon n}$$

$$\lesssim (LD\sqrt{\ln n} + B)\sqrt{\frac{\ln(1/\beta)}{n}} + LD\frac{[s\ln(1/\delta)\log(d/\beta)]^{1/4}}{\sqrt{\varepsilon n}}.$$

$\square$

### G.3 A Pure DP-ERM Algorithm for Nonsmooth Losses

We now prove that the rates of pure DP-ERM in the convex case above can be obtained without the smoothness assumption, albeit with an inefficient algorithm. This algorithm is based on the exponential mechanism, and it leverages the fact that the convex ERM with sparse gradient always has an approximate solution which is sparse. This result requires an additional assumption on the feasible set:

$$(x \in \mathcal{X} \wedge P \subseteq [d]) \implies x|_P \in \mathcal{X}, \tag{8}$$

where $x|_P \in \mathbb{R}^d$ is the vector such that $x_{P,j} = x_j$ if $j \in P$, and $x_{P,j} = 0$ otherwise. We will say that $\mathcal{X}$ is sparsifiable if (8) holds. Note this property holds e.g., for $\ell_p$-balls centered at the origin.

**Lemma G.2.** *Let $\mathcal{X}$ be a convex sparsifiable set. Consider the problem* (ERM) *under convexity (Item (A.3)), bounded diameter (Item (A.2)), Lipschitzness (Item (A.5)) and gradient sparsity (Item (A.7)), assumptions. If $x^*(S)$ is an optimal solution of* (ERM) *and $\tau > 0$, then there exists $\tilde{x} \in \mathcal{X}$ such that $\|\tilde{x}\|_0 \leq 1/\tau^2$, and*

$$F_S(\tilde{x}) - F_S(x^*(S)) \leq L\sqrt{s}\tau.$$

*Proof.* Let $\tilde{x} \in \mathbb{R}^d$ be defined as

$$\tilde{x}_j = \begin{cases} x_j & \text{if } |x^*_{S,j}| \geq \tau \\ 0 & \text{otherwise.} \end{cases}$$

Note that $\tilde{x} \in \mathcal{X}$ since $x^*(S) \in \mathcal{X}$ and $\mathcal{X}$ is sparsifiable. Now we note that

$$\|\tilde{x}\|_0 \leq \sum_{j:\, |x^*_{S,j}| \geq \tau} \frac{(x^*_{S,j})^2}{\tau^2} \leq \frac{1}{\tau^2}.$$

Finally, for the accuracy guarantee, we use convexity as follows,

$$\begin{aligned} F_S(\tilde{x}) - F_S(x^*(S)) &\leq \langle \nabla F_S(\tilde{x}), \hat{x} - x^*(S) \rangle \\ &\leq \|\nabla F_S(\tilde{x})\|_1 \|\tilde{x} - x^*(S)\|_\infty \\ &\leq L\sqrt{s}\tau, \end{aligned}$$

where in the last step we used that $\nabla f(\hat{x}, z_i) \in \mathcal{S}^d_s$ and the definition of $\tilde{x}$. $\square$

We present now the *sparse exponential mechanism*, which uses the result above to approximately solve (ERM) with nearly dimension-independent rates.

---

**Algorithm 7** `Sparse_Exponential_Mechanism`

---

**Require:** Dataset $S = \{z_1, \ldots, z_n\} \subseteq \mathcal{Z}$, $\varepsilon$ privacy parameter, $f(\cdot, z)$ $L$-Lipschitz convex function with $s$-sparse gradients and range bounded by $B$, $0 < \beta < 1$ confidence parameter

Let $\tau > 0$ be such that $\frac{\tau^3}{\ln(d/[\tau\beta])} = \frac{L\sqrt{s}\varepsilon n}{B}$

Let $\mathcal{N}_\tau$ be a $\tau$-net of $1/\tau^2$-sparse vectors over $\mathcal{X}$ with $|\mathcal{N}_\tau| \leq \binom{d}{1/\tau^2}\left(\frac{3}{\tau}\right)^{1/\tau^2}$

Let $\hat{x}$ be a random variable supported on $\mathcal{N}_\tau$ such that $\mathbb{P}[\hat{x} = x] \propto \exp\left\{-\frac{B}{\varepsilon n}F_S(x)\right\}$

**Return** $\hat{x}$

---

**Remark G.3.** *The bound on $|\mathcal{N}_\tau|$ claimed in Algorithm 7 follows from a standard combinatorial argument (e.g., [39]). Moreover, it follows that $|\mathcal{N}_\tau| \lesssim \left(\frac{d}{\tau}\right)^{1/\tau^2}$.*

**Theorem G.4.** *Let $\mathcal{X}$ be a convex sparsifiable set. Consider a problem (ERM) under bounded diameter (Item (A.2)), convexity (Item (A.3)), bounded range (Item (A.4)), Lipschitzness (Item (A.5)) and gradient sparsity (Item (A.7)), assumptions. Then Algorithm 7 satisfies with probability $1 - \beta$*

$$F_S(\hat{x}) - F_S(x^*(S)) \lesssim L^{2/3}B^{1/3}\left(\frac{s}{\varepsilon n}\ln\left(\frac{L\sqrt{s}\varepsilon n}{B}\frac{d}{\beta}\right)\right)^{1/3}.$$

*Proof.* Let $\tilde{x}$ be the vector whose existence is guaranteed by Theorem G.2. By the high probability guarantee of the exponential mechanism [1] with probability $1 - \beta$,

$$F_S(\hat{x}) - F_S(\tilde{x}) \leq \frac{B}{\varepsilon n}\left(\ln|\mathcal{N}_\tau| + \ln(1/\beta)\right) \lesssim \frac{B}{\varepsilon n}\frac{\ln\left(\frac{d}{\tau\beta}\right)}{\tau^2}.$$

Hence, using Theorem G.2 with the upper bound above,

$$\begin{aligned}
F_S(\hat{x}) - F_S(x^*(S)) &\leq F_S(\hat{x}) - F_S(\tilde{x}) + F_S(\tilde{x}) - F_S(x^*(S)) \\
&\lesssim \frac{B}{\varepsilon n}\frac{\ln(d/[\tau\beta])}{\tau^2} + L\sqrt{s}\tau \\
&\lesssim \left(L^2 B\frac{s}{\varepsilon n}\ln\left(\frac{L\sqrt{s}\varepsilon n}{B}\left(\frac{d}{\beta}\right)^3\right)\right)^{1/3},
\end{aligned}$$

where we used our choice of $\tau$. $\qquad\square$

