# OpenReview forum: "Differentially Private Optimization with Sparse Gradients"
_NeurIPS.cc/2024/Conference — NeurIPS 2024 poster_

### Official Review · Reviewer_L9WR · 2024-07-11

**Soundness:** 3
**Presentation:** 3
**Contribution:** 3
**Rating:** 7
**Confidence:** 3

**Summary:**

This paper explores differentially private optimization under the data/gradient sparsity assumption. For mean estimation with sparse data, it introduces new near-optimal bounds, improving previous results, especially in the high-dimensional setting. The corresponding lower bound is established using a novel block-diagonal construction. For DP ERM/SO with sparse gradients, the paper introduces a bias-reduced, randomly stopped SGD method, building upon their mean estimation mechanism. This approach achieves a nearly dimension-independent risk rate in the sparse gradient setting.

**Strengths:**

* The paper is technically solid, with results covering both ERM and SO. In addition, some results also include lower bounds.
* The novel block diagonal construction of lower bound and the analysis of randomly stopped noisy SGD will benefit future research.

**Weaknesses:**

The organization of the paper could be improved. For instance, adding more content to Section 6 and briefing the proof part in Section 5 would be beneficial.

**Questions:**

Can the authors provide the running times analysis for Algorithms 2 and 3?

**Limitations:**

The authors address the limitation of this work in the checklist

---

> ### Author Rebuttal · Authors · 2024-08-05
>
> Thank you for your positive assessment and feedback..
>
> In the revision, we will reorganize the content.  As you suggest, we will include results for DP-SCO from Section 6 (specifically from Appendix G.2), as well as a more extensive motivation and detailed proofs in Section 5.
>
> __Questions__
>
> Thank you for raising this question.  Regarding the running times, for Algorithm 2 we need to compute $2^{N+1}+1=O(n)$ gradients (where N is truncated geometric), and apply Gaussian mechanism and projection over the $\ell_1$-ball four times (these two algorithms run in time nearly linear in the dimension $d$). Overall, the gradient complexity is $O(n)$ and the computational complexity is $O(nd)$.
>
> For Algorithm 3, we can use Lemma 5.3 to provide an upper bound on the expected number of iterations (which is $O(n)$), and then multiply this number by the worst-case estimate of the complexity of Algorithm 2. This provides an in-expectation bound for the complexity of the algorithm.
>
> We will make sure to add these details in the revision.

---

> > ### Comment · Reviewer_L9WR · 2024-08-11
> > **Official comment by reviewer L9WR**
> >
> > Thank you for answering my question. I will keep my score.

---

### Official Review · Reviewer_GgyX · 2024-07-11

**Soundness:** 3
**Presentation:** 2
**Contribution:** 3
**Rating:** 5
**Confidence:** 4

**Summary:**

The paper studies differentially private optimization under the sparse gradient assumption. The paper first considers private and sparse mean estimation. Both lower-bounds and nearly matching upper bounds are established. The paper then use the mean estimation algorithm to construct private gradients in DP-SGD and obtain new dimension independent rates for both DP-ERM and DP-SCO under convex and non-convex settings of the objective function.

**Strengths:**

A crucial issue in differentially private optimization is that the utility rates depend on the dimension. This prevents the use of DP techniques to large models whose dimension scales to billions. The paper contributes to an important line of work in DP that tries to develop dimension free error rates under structual assumptions of the problem. Both the settings that gradients are sparse and the theoretical results under the sparsity assumption are novel and new.

**Weaknesses:**

1. There are several missing related works that should be included. [1-7] all derived dimension independent rates for DP optimization under various settings. [1] relaxed the dimension to a dependence on the rank of the feature matrix. [2, 5] studied the relaxed Lipschitz condition. [3,4] relaxed the dimension to a dependence on the trace of the Hessian. [6, 7] studied the semi-feature setting. Although they are not based on sparse gradients, I think it help to better position this paper when including all these existing dimension free rates in the DP optimization literature. Specifically, the assumption in [2, 5] is also only on the gradients, and [5] also develops an exponential mechanism that could be used for the pure DP case (though they did not discuss). In addition, [8] also studied multi-level Monte-Carlo for stochastic optimization.

2. All previous dimension free rates [1-5] are for the unconstrained case. As discussed in [1, 2], there seems to be a separation between the constrained and unconstrained settings. For unconstraiend case, with additional assumptions, both dimension free upper and lower bounds can be established. However, the dimension dependent lower bound in [Bassily, 2014] is constructed for a constrained generalized linear loss, which also applies to the case where gradients have additional structure. See more discussions in [2, 5]. Results in this paper seem to be contradictory to these related works, where dimension free rates exist for constrained optimization. Can authors explain why and provide more insights?

3. Other questions: (1) In the ideal case, the results for the sparse case should recover the non-sparse case when s=d. However, there seems to be a large gap; (2) In the algorithms, the knowledge of the sparsity s is required. How to determine s in practical applications and is it possible to design algorithms without knowing s? (3) Some assumptions overlap with each other. For example, if f is L-Lipschitz and domain is bounded by D, this already implies that $|f(x) - f(y)|\leq LD$; (4) In private optimization, we need to add noise to gradients, which makes gradients not sparse any more. How hard is it for the sparsity assumption to hold along the trajectory? Can sparsity be preserved for all x_t?

[1] Evading the Curse of Dimensionality in Unconstrained Private GLMs. AISTATS, 2021.

[2] When Does Differentially Private Learning Not Suffer in High Dimensions? NeurIPS, 2022.

[3] Dimension Independent Generalization of DP-SGD for Overparameterized Smooth Convex Optimization. arXiv, 2022.

[4] DPZero: Private Fine-Tuning of Language Models without Backpropagation. ICML, 2024.

[5] The Power of Sampling: Dimension-free Risk Bounds in Private ERM. arXiv, 2024.

[6] Deep learning with label differential privacy. NeurIPS, 2021.

[7] On Convex Optimization with Semi-Sensitive Features. COLT, 2024.

[8] On the Bias-Variance-Cost Tradeoff of Stochastic Optimization. NeurIPS, 2021.

**Questions:**

See weaknesses.

**Limitations:**

See weaknesses.

---

> ### Author Rebuttal · Authors · 2024-08-05
>
> Thank you for the valuable feedback.
>
> 1. Thank you for the references; we will make sure to add them to properly position our work within the field.
>
> 2. Unfortunately, it is not true that dimension-free rates are known only for unconstrained settings.  In fact, that there are works providing (nearly) dimension-free rates for the case of polytope feasible sets [22,23:Talwar, Thakurta, Zhang], [24: Asi, Feldman, Koren, Talwar] and [25: Bassily, Guzman, Nandi].  More importantly, there is _no contradiction with the lower bounds in BST14_, since their construction of fingerprinting codes (and packings for the pure case) uses _dense_ vectors. In fact, the key observation behind our sparse DP lower bound is that the sparsity constraint weakens this construction, and a block diagonal construction with blocks as in BST14 yields a nearly optimal lower bound. Note too that polynomial-in-the-dimension lower bounds are still obtained for large enough sample size $n$, showing a smooth transition between the different regimes.
>
> 3. Other questions
>
>     1. For mean estimation (Table 1), our upper bounds smoothly transition between the different regimes. For DP-SCO and DP-ERM (Table 2), it is also possible to obtain this transition (simply by selecting the algorithm with the best rate depending on the instance parameters). In the original table we decided to include only the new high-dimensional rates, but we have now added the different regimes (see the attached file). We apologize if this caused confusion.
>
>     2. Each specific application may have an ad-hoc way of estimating $s$. For example, in embedding models, the sparsity (for the embedding layer) will be the number of columns. Alternatively, $s$ can be a hyperparameter (this does not compromise privacy, as long as the hyperparameter selection is done privately).  Designing algorithms that work without knowing $s$ is an interesting question for future research---we will add this to the revision.
>
>    3. Regarding the overlap in Lipschitzness and boundedness assumptions, we opted to keep this (sometimes redundant) parameterization since we are combining different assumptions for different results. Thank you for pointing this out.
>
>    4. There might be a potential misunderstanding here; the gradient sparsity assumption is made over the raw _noise-free_ gradients. Any noise addition (or more general DP procedure) applied to these gradients does not break the sparsity assumption. Please let us know if this resolves your question.

---

> > ### Comment · Reviewer_GgyX · 2024-08-12
> >
> > Thanks for your response!
> >
> > I think a better understanding regarding the separation between constrained and unconstrained case mentioned in papers [1,2,5] is valuable to the community. Please check their arguments carefully and include a detailed discussion in the next version of the paper.
> >
> > In my understanding, if the gradients are always sparse and the support is fixed, then dimension-independent rates are natural regardless of the constrained sets as one can transform this d dimensional problem to an s dimensional one. However, if the support is not fixed, I am not clear what will happen. For the sparsity assumption, let me clarify my question. What I am confused is that $x_t$ can always be dense since we add noise to make it DP. Then how strong is this gradient sparsity assumption evaluated on every dense $x_t$?
> >
> > Anyway, I don't have other problems. I increase the score to 5, which is on the positive side. I am willing to support the acceptance of the paper.

---

> > > ### Author Response · Authors · 2024-08-13
> > > **Further responses**
> > >
> > > Thank you for engaging in further discussions. We proceed to answer your comments/questions.
> > >
> > > *[..] Please check their arguments carefully and include a detailed discussion in the next version of the paper.*
> > >
> > > We appreciate the provided references, and we will make sure to incorporate them in the next version.
> > >
> > > *In my understanding, if the gradients are always sparse and the support is fixed, then dimension-independent rates are natural regardless of the constrained sets as one can transform this d dimensional problem to an s dimensional one. However, if the support is not fixed, I am not clear what will happen. For the sparsity assumption, let me clarify my question. What I am confused is that $x_t$ can always be dense since we add noise to make it DP. Then how strong is this gradient sparsity assumption evaluated on every dense $x_t$?*
> > >
> > > We agree that the fixed support sparse gradient case is straightforward. We also agree that $x_t$ (and furthermore, its minibatch gradients) can be fully dense. However, this is not a contradiction as our assumption is imposed on the ***individual gradients at arbitrary points***. Now, if your question is directed to how strong is in practice to have sparse gradients at dense $x_t$, our answer is that  for our applications of interest is not strong. E.g. for embedding models the sparsity arises from the embedding of categorical features, hence it works regardless of the iterate. We hope this resolves your question.
> > >
> > > *[..] I increase the score to 5, which is on the positive side. I am willing to support the acceptance of the paper.*
> > >
> > > Thank you for taking our feedback into account.

---

### Official Review · Reviewer_JAyy · 2024-07-13

**Soundness:** 3
**Presentation:** 3
**Contribution:** 3
**Rating:** 6
**Confidence:** 3

**Summary:**

The paper provide algorithms for DP optimization with sparse gradient, proving both upper bounds and lower bounds, which are almost match.

**Strengths:**

1. The paper has a nice presentation, starting from sparse mean estimation upper bounds and lower bounds and then go into ERM with sparse gradients and deal with bias issues introduced by the projection estimator.
2. The upper bounds and lower bounds almost match.

**Weaknesses:**

1. It is a pure theory paper, it would be more interesting to see if the algorithm can inspire improved practical applications.
2. The content is a bit dense, without summary or conclusion sections.

**Questions:**

The proposed algorithms seems not too hard to use in practice, could the authors comment whether they can be used practically? Or if the authors tested them in practice, it would be interesting to see some results.

**Limitations:**

Limitations are well covered in the content.

---

> ### Author Rebuttal · Authors · 2024-08-05
>
> Thank you for your review. We will now elaborate on the comments.
>
> Thank you for your comment on practical applications. In this regard, the idea of gradient sparsity has already impacted practical DP optimization. The most immediate references in this respect are [6: Zhang, Mironov, Hejazinia] and [7: Ghazi, Huang, Kamath, Kumar, Manurangsi]. Regarding the specific contributions of this work, we believe that significant components of our results can inspire practical improvements. First, the projection mechanism and compressed sensing approaches are easy to apply and should lead to better statistical performance in mean estimation than, e.g., the sparse vector technique (which is the main approach in [6,7]).  Second, our regularized output perturbation approaches for DP-SCO (Section G) are easy to implement.  Third, towards neural network applications (such as embedding models), the computational benefits of sparsity are more significant under structured sparsity, e.g., in the case of embedding models we need the sparsity to operate at the level of rows of the embedding matrix. We believe that ideas used in the context of group or structured sparsity can be of use here, but this is out of the scope of the current submission. Finally, a major roadblock for practical applications is the heavy-tailed nature of the bias-reduced gradient estimator used for SGD. While the algorithm in its current form might be less practical, we hope our work may inspire further ideas and lead to more practical methods.
>
> We apologize for the current dense content; in the revision, we will try to rewrite to make it more accessible.
>
> __Questions__
>
> As we mention above, the bias reduction method may be hard to implement in practice. Particularly, the heavy-tailed nature of the gradient estimator may pose convergence challenges. Boosting might mitigate this, but we are interested in continuing investigating alternative approaches that can be more practical.

---

> > ### Comment · Reviewer_JAyy · 2024-08-12
> >
> > Thank you for the response. I will keep my score.

---

### Official Review · Reviewer_9vLf · 2024-07-15

**Soundness:** 3
**Presentation:** 2
**Contribution:** 3
**Rating:** 6
**Confidence:** 3

**Summary:**

This paper addresses the problem of DP-Convex optimization and DP-SCO in scenarios where the gradient of each individual sample is $s$ sparse. The main question explored is how sparsity can help improve the known rates for DP convex optimization.

The main contributions of the paper are as follows:

DP Sparse Mean Estimation of High-Dimensional Vectors: The authors investigate DP mean estimation under the assumption of sparsity. They use the projection mechanism from Nikolov et al. to propose algorithms for both pure-DP and approximate-DP settings. The error of the algorithm scales with  $s$ when the number of samples is moderate. The main results are presented in Theorems 3.2 and 3.3.

Lower Bound for Mean Estimation under Sparsity: The authors provide lower bounds for mean estimation under sparsity. Note that due to the reduction in BST14, the lower bound on mean estimation can be translated to a lower bound on DP-ERM.

Algorithms for DP-ERM under Sparsity: The authors propose an interesting algorithms for DP-ERM under the sparsity assumption. To compute gradients, they use their proposed algorithm for mean estimation as a black-box. Unlike DP-SGD, the gradient in this case is "biased." To address this issue, the authors introduce an interesting approach using random batch sizes with an exponentially increasing schedule to achieve a bias similar to that of a full-batch algorithm.

**Strengths:**

I think the paper addresses an important problem with practical relevance. I think the paper is complete in the sense that the authors provide near complete story of optimization under sparsity at least in terms of achievable rates. The idea of behind the optimization algorithm is interesting.

**Weaknesses:**

The main drawback of this work is its presentation. Some of the proof is very difficult to parse. I have difficulty understanding the many algorithmic choices behind Algorithm 2. In particular, the idea of using random batch size, fully adaptive DP mechanism, specific distribution for batch size are not clear to me. I think the authors should provide a discussion on the necessity of these particular algorithmic choices.

**Questions:**

I can't fine the results regarding DP-SCO in the paper. Maybe I am missing something in the paper.

1- Is there any way to extend the results to the case with approximate sparsity?

2- what is p_n in algorithm 2?

3- What is the importance of random batch size?

4- Theorem A.6 seems to have a typo?

5- Line 641 to 642, 1[T>=t] has been changed to 1[T<=t]. It is not clear to me? Step 1 needs more clear explanation.

6- What is the shortcoming of “equal” privacy budget allocation as in the BST14 paper?

7- What is the issue with using full-batch for computing the gradient?

8- What is the definition of $\mathcal{F}_{t-1}$ in prop. E.1?

9- Line 644, it is not clear to me why conditional expectation of bias and variance are also bounded by b and v.

---

> ### Author Rebuttal · Authors · 2024-08-05
>
> We would like to thank the reviewer for the valuable and detailed feedback.
>
> Firstly, we apologize for the lack of clarity of the algorithmic choices and the proofs in our submission.  Here is an overview of the choices behind Algorithm 2.  As discussed in the paper, the near-optimal mean estimation algorithms we introduce are _biased_. Simply using SGD with this mean estimation procedure for minibatch gradients would result in a _polynomial degradation_ of the rates (see, e.g., [6: Zhang, Mironov and Hejazinia], who carry out this approach). To handle this issue, we propose a _bias-reduction method_, which closely follows the approach in [14: Blanchet and Glynn]. This randomized method produces an estimator that telescopes in expectation, resulting in bias which scales as the one of the largest possible batch, but with minibatch sizes that are typically much smaller. The exponential range of the batch size follows from the telescoping idea.
>
> The bias-reduction approach is beneficial for privacy, because when minibatches are smaller, one can leverage _privacy amplification by subsampling_. However, since batch sizes are random, the classical advanced composition DP analysis does not suffice. Here is where we use the _fully adaptive composition_ theorem [15: Whitehouse, Ramdas, Rogers and Wu]: in particular, since our batch randomization is predictable, we can define stopping times in such a way that we do not exceed the privacy budget. This however introduces challenges in the SGD analysis, which are addressed by the _boosting_ procedure.
>
> We will include a better detailed overview of this algorithm and its analysis in the revision.
>
> __Questions__
>
> About DP-SCO results. Due to space limitations, all results for DP-SCO were deferred to Appendix G. In the final version of the paper we will include the main results from that section (particularly, Appendix G.2).
>
> 1. Yes. As pointed out in Remark 2.1, in all of our upper bounds we can replace the set of sparse vectors by a scaled $\ell_1$-ball. This set is a more robust way to quantify sparsity, as known from the compressed sensing literature.
>
> 2. $p_n$ is the probability of choosing a batch size $2^{n+1}$, i.e., $p_n=C_M/2^n$ (where $C_M$ is a normalizing constant). This is explained in lines 233-236, but we will make it explicit in the pseudocode to avoid confusion.
>
> 3. Random batch sizes allow bias reduction with smaller minibatches, which are amenable for privacy amplification by subsampling. Using a full batch estimator would incur similar bias, but no privacy amplification. We will make sure to expand on this important aspect.
>
> 4. Thank you. We found an incorrect indexing: $[1:t-1]$ instead of the correct $[0:t-1]$.
>
> 5. Apologies for causing confusion: we simply wrote the same event in two different ways, i.e.,  $\{t\leq T\}$ and $\{T\geq t\}$. In the revision, we will retain the same notation.
> In the first step we use the regret bound for biased SGD, and in the second step we write the finite sum from $0$ to $T$ as a series with zero terms (due to the indicator) for $t>T$. Please let us know if more clarification is needed.
>
> 6. We are not sure which part of the paper you are referring to. Can you please let us know, and we can clarify it accordingly?
>
> 7. Please see answer 3 above.
>
> 8. $\\{ \cal F_t \\}_{t\in\mathbb{N}}$ is the natural filtration, i.e., ${\cal F}_t = \sigma((x^s), {s \leq t})$. We will add a clarification of this in the revision.
>
> 9. The sources of randomness used in Lemma 5.2 are only the batch size r.v. N and the DP noise used to compute ${\cal G}(x)$. While we could have stated Lemma 5.2 for an iterate $x^t$ (and correspondingly condition on ${\cal F}_{t-1}$), we believe the current presentation of this lemma is cleaner, and avoids unnecessary notation.

---

> > ### Comment · Reviewer_9vLf · 2024-08-11
> >
> > Thanks for the detailed response!
> >
> > Regarding question 6 in my initial review: the "usual" analysis of DPGD is based on the following: assume we want to run the algorithm for $T$ iterations and the privacy budget is $\rho$-zCDP. Then, we set the noise variance such that the privacy budget at each iteration is $\rho/T$. I want to better understand the shortcomings of this approach compared to the proposed method in the paper.

---

> > > ### Author Response · Authors · 2024-08-11
> > > **Answer to question 6 by reviewer 9vLf**
> > >
> > > Thank you for the clarification. Before answering, please keep in mind that for Noisy SGD algorithms the minibatch gradients estimators are unbiased, so what we discuss next is specific to the sparse setting.
> > >
> > > In the sparse setting we only have at our disposal (minibatch gradient) mean estimation algorithms whose bias scales as $1/\sqrt{B}$ where $B$ is the batch size. Our main limitation then comes from this bias (which does not vanish with a larger number of iterations); moreover, note that the benefits of privacy amplification by subsampling take place for smaller values of $B$. Hence these two effects pose a strong privacy/accuracy tradeoff.
> > >
> > > Regardless of the batch size, and as you correctly point out, we also need to take into account the effect of composition. Optimizing on both batch size and number of steps, one can see that the biased SGD algorithm will converge with a rate which has polynomial gaps  in $1/\varepsilon$ and $s$ (compared to the lower bounds); see, e.g. the work of Zhang, Hejazinia and Mironov [6]. By contrast, our bias reduction approach leads to smaller bias but can still benefit from privacy amplification by subsampling. This leads to various technical challenges which are the core of our analysis in Section 5.
> > >
> > > We hope this resolves your question and allows you to better appreciate our technical contributions.

---

### Author Rebuttal · Authors · 2024-08-06

Updated table incorporating the low and high dimensional rates for DP optimization.

---

### Decision · Program_Chairs · 2024-09-25

**Decision:**

Accept (poster)

**Comment:**

Pros:

- address the important problems of DP mean estimation and optimization in the natural setting where each individual point/gradient is guaranteed to be sparse.

- the rate for mean estimation is nearly complete with upper and lower bounds that are close

Cons:

- the current rates for optimization in the sparse case are polynomially different from the rates for the general dense case and it is not clear if they are optimal.

- the assumption of exact sparsity can be restrictive. Several reviewers suggest addressing the approximately sparse setting.

The paper's formatting deviates from the template (the page size is 9.68in by 11in). Please fix this issue.